

# Emissions of biogenic volatile organic compounds from agricultural lands and the impact of land-use and other management practices: A review

Yang Liu[1,*], Raluca Ciuraru[1], Letizia Abis[2], Crist Amelynck[3,4], Pauline Buysse[1,+], Alex Guenther[5], Bernard Heinesch[6], Florence Lafouge[1], Florent Levavasseur[1], Benjamin Loubet[1], Auriane Voyard[1], and Raia-Silvia Massad[1]

[1]Université Paris-Saclay, INRAE, AgroParisTech, UMR ECOSYS, 91120 Palaiseau, France
[2]Umweltchemie und Luftrinhaltunz, Technische Universität Berlin, 10623 Berlin, Germany
[3]Royal Belgian Institute for Space Aeronomy, 1180 Brussels, Belgium
[4]Department of Chemistry, Ghent University, 9000 Ghent, Belgium
[5]Department of Earth System Science, University of California, Irvine, 92697 Irvine, California, USA
[6]Gembloux Agro-Bio Tech, Université de Liège, 5030 Gembloux, Belgium
[+]now at UMR SAS, INRAE - Institut Agro Rennes-Angers, 35000 Rennes, France

**Correspondence:** Yang Liu (yang.liu@inrae.fr)

**Abstract.** Crops and soils are essential sources of biogenic volatile organic compounds (BVOCs) from the agriculture landscape. Agricultural management practices, including species choice, tillage, fertilization, irrigation, and cover crop application, affect soil nutrient levels, crop growth, microbial density and activities, and trigger changes in BVOC emission rates from both crop and soil. A better comprehension of the emission processes and controlling factors can improve model representation, reduce uncertainties, and allow more accurate quantitative estimations of agricultural BVOC emissions. We summarized current knowledge on BVOC emissions from common agricultural crops (oilseed rape, wheat, maize), cover crops, and bare soil under different management practices. The current challenges for improving the representation of agricultural BVOC emissions in models and a conceptual model for estimating BVOC emissions from agricultural land surfaces are discussed.

## 1 Introduction

Biogenic volatile organic compounds (BVOCs) are primarily emitted by vegetation during plant growth, reproduction, and as a stress protection mechanism (Andersson et al., 2002; Faiola and Taipale, 2020). Globally, BVOCs contribute significantly to atmospheric volatile organic compound (VOC) concentrations and are generally more reactive than anthropogenic VOCs (Calfapietra et al., 2013; Havermann et al., 2022). With a decline in anthropogenic VOC emissions in most developed regions, BVOCs now dominate total VOC reactivity on regional scales, even in some urban areas (Gu et al., 2021). Rapid oxidation by atmospheric oxidants, such as hydroxyl radical ($OH$), nitrate radical ($NO_3$), and ozone ($O_3$), makes BVOCs integral to atmospheric chemistry processes (e.g. Ng et al., 2017), influencing local or regional air quality and climate (Peñuelas and Staudt, 2010; Kulmala et al., 2012).



Agricultural BVOC emissions can significantly impact atmospheric chemistry, affecting air quality and human health, especially in regions with a high proportion of agricultural fields surrounding populated areas (Food and Agriculture Organization,

2023, https://data.worldbank.org/indicator/AG.LND.AGRI.ZS, last time access: 25/04/2023). Improving the detection and understanding of agricultural BVOCs could enhance strategies for pest defense, soil fertility recovery (Brilli et al., 2019), and increased yield production to meet food security and bioenergy development needs (Fujimori et al., 2022; Viana et al., 2022). Given the magnitude and chemical composition of BVOCs, along with variations related to multiple conditions such as species, local/regional environments, and management practices, comprehensive agricultural emission measurements are essential.

Previous agricultural BVOC studies have primarily focused on grain species (e.g., wheat, maize, rye, and corn), vegetables (e.g., potato, tomato, and cucumber), oilseed plants (e.g., oilseed rape and canola), and fruit plantations (e.g. Gentner et al., 2014). However, differences in reported characteristics among studies pose challenges for synthesizing observations. Factors such as plant phenological stages, soil composition, and microbial activity contribute to variations in BVOC emissions. In addition to plants, the soil layer is a potential source and sink of BVOCs, originating from microbial activity and root respira-

tion, as well as physico-chemical processes at soil-water interfaces (Insam and Seewald, 2010; Ruiz et al., 1998; Tang et al., 2019). Soil BVOC emissions are influenced by factors like temperature, moisture conditions, microbial metabolism, and soil properties, highlighting the complexity of the interactions involved.

Agricultural management practices play a pivotal role in altering BVOC emissions from crops and soil (Kumar et al., 2011). Choices such as crop species rotation, cover crops, and fertilization impact both emission capacity and soil properties, affecting

nutrient levels and water balance (Potard et al., 2017; Haider et al., 2022; Hirzel et al., 2021). Tillage and irrigation further influence soil conditions and BVOC exchange rates (Scagel et al., 2011; Mateo-Marín et al., 2022). Climate interacts with these management practices, introducing additional challenges in estimating and predicting BVOC emissions from agricultural landscapes (Agelopoulos et al., 2000; Loreto and Schnitzler, 2010).

This study focuses on understanding the processes behind agricultural BVOC emissions, with objectives including a review

of existing studies, identification of processes responding to different phenological stages and agricultural management practices, and summarization of the status and challenges in agricultural BVOC model development. Notably, this review excludes BVOCs related to pest attacks and semi-volatiles released from pesticides, which have been addressed elsewhere (Faiola and Taipale, 2020; Yusà et al., 2009). Sections 2 and 3 of the paper provide a structured review of existing studies around different management practices and crop phenologies, followed by a discussion on challenges in agricultural BVOC modeling (Sect.4)

and a proposed conceptual model for emissions from plants and soils (Sect.5).

## 2  Emissions of BVOCs from bare soil, different crop phenological stages and roots

For agricultural fields, BVOCs can be emitted by all organs of crops above and below ground during the different plant developmental stages. The most emitted compounds and compound groups depend on the phenological stage of crops (Manco et al., 2021; Havermann et al., 2022), mainly due to changes in plant development and maintenance processes (Baggesen et al.,

2021; Folkers et al., 2008). Considering that soil is another important emitter for the total agricultural BVOC budget (Tang



et al., 2019), we summarize the articles focusing on crop BVOC emissions in different phenology stages (Table.1) and bare soil in this section. Crop phenological stages include tillering, stem elongation, heading and flowering, and ripening.

**Table 1.** Crop BVOC emission during different phenology stages ($\mu$g $m^{-2}(leaf)$ h$^{-1}$)

| Compounds | Species | Tillering to Inflorescence emergence | Heading/Flowering | Ripening |
|---|---|---|---|---|
| Methanol | Maize | 92[a,R7], 12.2[*,R6], 5.0[b,R7] | 135[i,R1], 5.9[c,R7], 2580[i,R3] | 69.1[R1], 42.5[d,*,R7] |
| | Oilseed rape | 900.2[e,R1] | 3902[i,R1] | 2550[d,R3], 5970[d,end,R3] |
| | Wheat | -17.2[f,R4], 40.9[g,R4], 63.5[g,R4] | 527[j,R5] | 900[R6], 144.5[d,R4], 232.5[d,R5] |
| | Ryegrass | | 1648[h,R1], 558[i,R1] | |
| Acetaldehyde | Maize | 0.2[*,R6] | 97.9[i,R1] | 56.7[R1], 12[d,*,R7] |
| | Oilseed rape | 17.9[e,R1] | 118[i,R1] | 40[d,end,R3] |
| | Wheat | -11.6[f,R4], -1.6[g,R4], -2.4[g,R4] | 59[j,R5] | 3.5[R6], 18.6[d,R4], 30[d,R5] |
| | Ryegrass | | 185[h,R1], 58.5[i,R1] | |
| Ethanol | Maize | | 12.6[i,R1] | 16.4[R1] |
| | Oilseed rape | 5.9[e,R1] | 109[i,R1] | 210[d,R3], 110[d,end, R3] |
| | Wheat | | 240[j,R5] | |
| | Ryegrass | | 48[h,R1], 4.3[i,R1] | |
| Acetone | Maize | 1.2[*,R6] | 95.4[i,R1] | 71.1[R1] |
| | Oilseed rape | 20.1[e,R1] | 87.4[i,R1], 40[i,R3] | 40[d,R3], 80[d,end,R3] |
| | Wheat | -2.8[g,R4], -1.3[g,R4] | 43[j,R5] | 9.4[R6], 16.4[d,R4], 130[d,R5] |
| | Ryegrass | | 92[h,R1], 71.7[i,R1] | |
| Acetic acid | Maize | | | 9.6[d,*,R7] |
| | Oilseed rape | 9.3[e,R1], 78[e,R2] | 208[i,R1], 400[i,R2], 20[i,R3] | -20[d,R3], -30[d,end,R3] |
| | Wheat | -7.7[f,R4], 3.1[g,R4], -15.1[g,R4] | | -19.6[d,R4], -16.1[d,R5] |
| | Ryegrass | | 6.1[h,R1] | |
| Isoprene | Maize | 0.2[*,R6] | | |
| | Oilseed rape | 8.5[e,R1] | 31[i,R1], 10[i,R3] | 10[d,R3] |
| | Ryegrass | | 17.1[h,R1], 8.2[i, R1] | |
| MACR+MVK | Oilseed rape | 2.8[e,R1] | 24[i,R1] | -10[d,R3] |
| | Ryegrass | | 6.2[h,R1], 0.4[i,R1] | |
| Toluene | Oilseed rape | 0.5[e,R1] | 10.4[i,R1] | |



| | | | | |
|---|---|---|---|---|
| Hexenal | Maize | | 34.1[i,R1] | 26.5[R1], 8[d,*,R7] |
| | Oilseed rape | 1[e,R1] | 16.2[i,R1] | |
| | Ryegrass | | 11.3[h,R1], 1.3[i,R1] | |
| Hexanal | Maize | | 83.9[i,R1] | 65.6[R1] |
| | Oilseed rape | 3.2[e,R1] | 76.3[i,R1] | |
| | Ryegrass | | 13[h,R1], 0.7[i,R1] | |
| Xylenes | Maize | | 9.6[i,R1] | 9.3[R1] |
| | Oilseed rape | 0.5[e,R1] | 8.8[i,R1] | |
| Junipene | Oilseed rape | 1.5[e,R1] | 1.3[i,R1] | |
| Formaldehyde | Oilseed rape | | -40[i,R3] | 670[d,end,R3] |
| Formic acid | Oilseed rape | | 20[i,R3] | -90[d,R3], -60[d,end,R3] |
| Methanethiol | Oilseed rape | | | 20[d,R3], 40[d,end,R3] |
| | Wheat | | 0.8[j,R5] | |
| Methyl salicylate | Oilseed rape | 1[e,R2] | 0.2[e,R2] | |
| Total Monoterpenes | Maize | 6.7[a,R7], 0.2[*,R6], 0.7[b,R7] | 0.5[c,R7] | |
| | Oilseed rape | 0.4[e,R1] | 50[i,R3] | 40[d,R3] |
| | Wheat | -2.0[f,R4], 1.7[g,R4], -3.6[g,R4] | | 5.3[d,R4], 1.4[d,R5] |
| α-Pinene | Oilseed rape | 70[e,R2] | 29.8[e,R2] | |
| Camphene | Oilseed rape | 4.3[e,R2] | 2.5[e,R2] | |
| 3-Carene | Oilseed rape | 86[e,R2] | 35[e,R2] | |
| β-Pinene | Oilseed rape | 19[e,R2] | 7.5[e,R2] | |
| Limonene | Maize | | 132[i,R1] | 48.4[R1] |
| | Oilseed rape | 15[e,R2] | 9.1[e,R2] | |
| 1,8-Cineole | Maize | | | 4.4[R1] |
| | Oilseed rape | | 9.3[i,R1] | |
| Total Sesquiterpenes | Oilseed rape | 1.5[e,R1] | | |
| α-Farnesene | Oilseed rape | 1[e,R2] | 0.7[e,R2] | |
| α-Humulene | Maize | | 28.4[i,R1] | 5.7[R1] |
| Isolongifolene | Ryegrass | | 2.5[h,R1], 4.4[i,R1] | |

Note: [a]Young leave, [b]Semi-mature leaves, [c]Mature leaves, [d]Senescent leaves, [e]Inflorescence emergence; [f]Leaf unfolding; [g]Ear formation; [h]Heading; [i]Flowering; [j]Leaf development; [*]Maximum value. MACR: methacrolein, MVK: methyl vinyl ketone; R1: Havermann et al., 2022; R2: Veromann et al., 2013; R3: Flux measurements by the Integrated Carbon Observation System (ICOS) at Grignon station in France in 2017 (Buysse et al., 2024); R4: Bachy et al., 2020; R5: Loubet et al., 2022; R6: Gonzaga Gomeza, et al., 2019, converted by multiplying the specific leaf weight (1000 g m$^{-2}$, Bachy et al., 2020); R7: Mozaffar et al., 2018, presented initially in $\mu$g gdw$^{-1}$ h$^{-1}$ and converted by multiplying the specific leaf weight (0.062 m$^2$ of leaf per gram for young leaves, 0.034 m$^2$ of leaf per gram for mature and semi-mature leaves, and 0.052 m$^2$ of leaf per gram for senescent leaves, personal communication with Crist Amelynck).



## 2.1 Bare soil

The agricultural soil primarily emit BVOCs through microbial metabolism, especially under a warm condition, while soil
particles and soil water could also desorb BVOCs, thereby increasing their emissions (Schade and Custer, 2004; Asensio et al.,
2008a; Ramirez et al., 2010). Soil microbes absorb some BVOC and contribute to the overall net emission (Tang et al., 2019;
Abis et al., 2020). Generally, emissions from bare soil are less than vegetation and are dominated by methanol and acetone,
followed by acetaldehyde and acetic acid (Mancuso et al., 2015; Schade and Goldstein, 2001) (Table 2). However, acetone
emissions may be the highest in this stage (bare soil) compared to the growing plant season (Bachy et al., 2016). A short-time
emission peak may appear, especially after tillage and sowing (Rossabi et al., 2018). Terpenes are produced in the soil as a
microbial metabolite, but their emissions are mostly negligible when compared to total atmospheric composition (Horváth
et al., 2012; Rossabi et al., 2018).

**Table 2.** BVOC emission from soil (unit in $\mu$g m$^{-2}$ h$^{-1}$)

| Soil type | Loam | Loamy sand | Silt loam | Silt clay[1] | Silt clay[2] |
|---|---|---|---|---|---|
| Methanol | 533–2933 | 0–533 | 335 | 0.05 | 214.08 |
| Acetaldehyde | n.a. | n.a. | 102 | -0.05 | 9.04 |
| Acetone | 81–258 | -16–81 | 136 | 0.63 | n.a. |
| Acetic acid | n.a. | n.a. | n.a. | 0.22 | 2.16 |
| Ethanol | n.a. | n.a. | n.a. | n.a. | 41.13 |
| Isoprene | n.a. | n.a. | n.a. | n.a. | 1.47 |
| Monoterpene | n.a. | n.a. | n.a. | 0.02 | n.a. |
| Soil state | With litter | Bare soil | Bare soil | Bare soil | With residues |
| Vegetation, Country | Forest, United States | Agriculture, Germany | Agriculture, Belgium | Agriculture, Italy | Agriculture, France |
| Reference | Schade and Goldstein (2001) | Schade and Custer (2004) | Bachy et al. (2016) | Manco et al. (2021) | Grignon station[a] |

Note: [a]Flux measurements at Grignon station in France in 2017. n.a.: No data available. Silt clay[1]: 52% of clay, 28% of silt, 20% of sand. Silt clay[2]: 25% of clay, 70% of silt, and 5% of sand (with rapeseed residues).

## 2.2 Leaf growth and senescence

Emissions of BVOCs from leaf vary among species and phenological stages. For example, methanol is the dominant compound
released during leaf growth for most crops (Mozaffar et al., 2018; Gonzaga-Gomez et al., 2019), and its emissions are affected
mainly by cell wall changes during leaf development (MacDonald and Fall, 1993). Short-chain aldehydes (e.g., formaldehyde,
acetaldehyde) and sulfur compounds also vary with leaf development (Seco et al., 2007; Danner et al., 2015). For tomato
leaves, a high amount of $\alpha$-phellandrene was reported (Andersson et al., 1980), which may be emitted from external/internal
storage organs like glandular trichomes (Tissier et al., 2017; Kortbeek et al., 2016), and for grape leaves, one of the main





emitted compound is butanone (Ilc et al., 2016), but these two compounds are seldom detected in other crops. Isoprene, which is not stored in plants but produced from recent photosynthesis products (Guenther et al., 1995) is emitted at relatively low rates for crops. Loubet et al. (2022) measured negligible net isoprene emissions from a wheat field in France, which may be caused by the phenology-specific emission pattern because the low isoprene emission was also detected by enclosure measurement during wheat leaf senescence (Gonzaga-Gomez et al., 2021).

Leaf age also affects the composition and abundance of BVOC emitted from plants (Guenther et al., 2012; Gonzaga-Gomez et al., 2021), with their emissions decreasing with leaf aging (Morrison et al., 2016; Abis et al., 2021). MacDonald and Fall (1993) stated that younger tomatoes, cucumbers, and soybean leaves can emit 2 to 8 times higher BVOCs than mature leaves. The reason for larger emissions from younger leaves relates to the Pectin Methyl Esterases during cell wall remodeling, as well as the net photosynthesis rate being highest for young leaf (Mozaffar et al., 2017; Song et al., 2023). However, Bachy

et al. (2020) and Gonzaga-Gomez et al. (2021) both show that young leaves emit less methanol for wheat. Besides methanol, sesquiterpenes may increase with the increasing leaf age (Huang et al., 2015), but this response has only been detected in woody plants and needs to be investigated in agricultural herbaceous plants. However, the leaf aging effect was investigated at the leaf scale, BVOC emissions may be underestimated when leaf biomass reaches its maximum on an ecosystem scale (Baggesen et al., 2022).

**2.3    Flower and fruit**

Phenology stages for the flowering stage include inflorescence emergency, heading, and flowering, while fruit development and ripening are fruit phenologies. Methanol and acetaldehyde are dominant compounds during flower stages according to measurements on maize, and oilseed rape (Veromann et al., 2013; Havermann et al., 2022), which is similar to the measurements on tundra vegetation in the corresponding phenology period (Baggesen et al., 2021). A considerably high emission rate of

monoterpenes is measured during the inflorescence emergence of oilseed rape, including the emission of $\alpha$-pinene, $\beta$-pinene, and camphene that were higher than in the following stages (Veromann et al., 2013). In the flowering stage, the $\alpha$-thujene starts to be emitted, and the emission of acetic acid is higher than during the bud stage (Veromann et al., 2013). Concerning maize, for example, emissions are about two times higher in flower stages than other phenological stages for methanol, limonene, and green leaf volatiles (Havermann et al., 2022). It is important to note that crops are more sensitive to environmental conditions

during flower stages, especially to changes in temperature and moisture, which can largely be affected by climate change, and impact BVOC emissions. The soil continues emitting BVOCs during plant growth and ripening, but emission rates from this period have not been reported so far to our knowledge.

**2.4    Participation of roots in agricultural BVOC emissions**

Characterization of root contents and measurements taken close to the roots or in the absence of soil tend to confirm the

capacity of crop plant roots to emit VOCs (La Forgia et al., 2020; Gulati et al., 2020; Lee Díaz et al., 2022). However, roots modify the physico-chemical properties of the soil, e.g., its porosity, pH, organic matter content, and moisture (Nye, 1981; Angers and Caron, 1998). These changes can affect the rate of VOC transport via diffusion in the gaseous phase of the soil and





the VOC temporary retention via dissolution in the aqueous phase and adsorption at the gas-liquid and liquid-solid interfaces (Peñuelas et al., 2014; Insam and Seewald, 2010; Ahn et al., 2020). Moreover, the root system can hardly be distinguished
from the surrounding rhizosphere, i.e. the soil volume in direct contact with the roots and of unique microbial richness. By selecting and promoting the development of certain microbial communities in the soil, the roots influence the overall role of soil microorganisms as a sink or source of VOCs. For example, except in rice paddies, the amplitude of methanol oxidation and methylotrophy by bacteria is higher in the rhizosphere of crops than in bulk soil (Ling et al., 2022). As a result, besides the fate of VOCs in soil is highly related to their physico-chemical properties, the evaluation of the sink or source character
of roots for VOCs remains complex and can vary depending on the environmental conditions, and the experimental protocol (Tang et al., 2019).

The scarce studies aiming to comprehensively characterize the stress-free fluxes of VOCs at the soil surface in the presence of roots have been limited to forest trees and have shown that roots can be either a VOC sink or source (Asensio et al., 2008b; Trowbridge et al., 2020; Gray et al., 2014). However, the magnitude of these fluxes at the soil surface remains low compared
to VOC emissions from the shoots. Similar studies on crop species are missing since most studies with crops focused on stress-induced root VOC emissions without having quantified the overall VOC root fluxes. Some VOCs produced by crop roots are likely constitutively emitted, unrelated to stress, and might be more representative of certain crop species. For example, rapeseed roots emit sulfur compounds such as methanethiol, dimethyl sulfide, and dimethyl disulfide (Acton et al., 2018; van Dam et al., 2012; Danner et al., 2012). Other studies even showed that root VOC emissions differed among the genetic
backgrounds of the same crop plant. For example, La Forgia et al. (2020) found higher VOC root content from one maize variety compared to another, explained mainly by hexanal, heptanal, and 2,3-octanedione root content. Lee Díaz et al. (2022) detected five (3-nonene, heptanal, 2-carene, 1-(2-hydroxyphenyl)-ethanone and methyl salicylate) and three (camphene, $\beta$-pinene, and $\gamma$-terpinene) VOCs from non-stressed root emissions of domesticated and wild tomato species respectively, that were not emitted by the other one.

## 3 Effects of agricultural management practices on BVOC emission

BVOC emission is affected by agricultural management practices, including species choice (crop, cover crop, and grass), fertilization (mineral and organic), irrigation, tillage, and residue management, which are discussed in this section. These human-induced changes to the agricultural land impact the surfaces' physicochemical and biological properties, potentially altering BVOC emissions (Fig.1). A summary of common management practices and their impacts on soil-plant variables is
presented in (Table A1).

Studies show that methanol, acetone, and acetaldehyde are dominant compounds emitted from bare soil and over the entire crop growing season under different agricultural management practices, while other compounds with lower emission rates could be emitted during specific management practices, e.g., ethanol, dimethyl sulfide, formaldehyde, isoprene, etc. (Table 3).




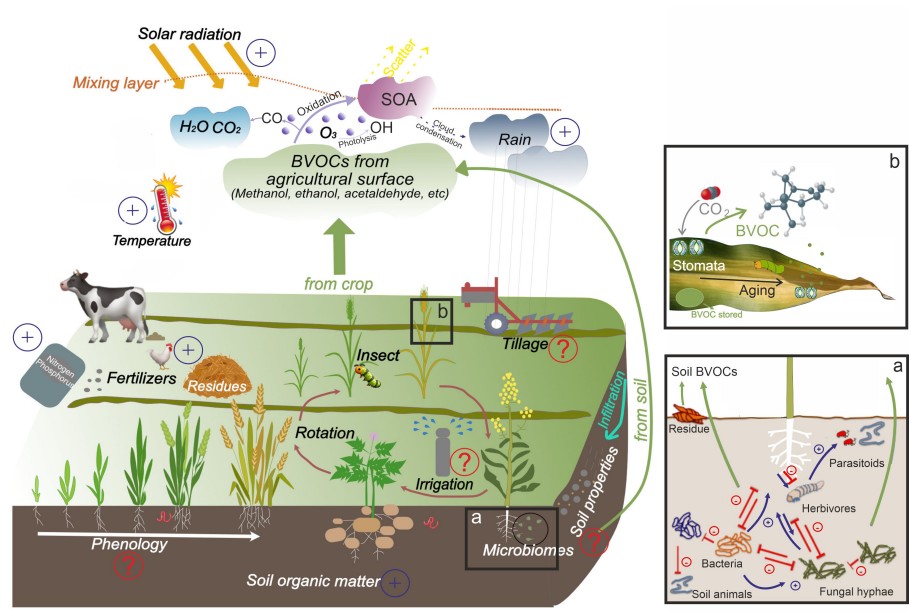

**Figure 1.** Scheme of BVOC emissions under from agricultural including management practices (left). (a) Soil BVOC emitting processes (Modified based on (Peñuelas et al., 2014)). (b) BVOC emitting processes on leaf scale (Modified based on (Mozaffar et al., 2017; Huang et al., 2018)). The source of agricultural BVOCs are presented in green arrows, Blue arrows, and blue circles with cross present factors that possibly positively impact BVOC emissions. The red arrow and the red circle with a minus symbol present factors that may negatively affect BVOC emission. The circle with a question mark represents processes where no apparent a negative/positive effects has been reported from previous studies. SOA: secondary organic aerosols.





**Table 3.** Current process understanding of BVOC emission from agriculture ecosystem

| Compounds | Chemical formula | Species choice in rotation | Mineral fertilization | Organic fertilization | Irrigation | Tillage |
|---|---|---|---|---|---|---|
| Methanol | $CH_4O$ | ++ | n.a. | ++ | n.a. | ++ |
| Acetaldehyde | $C_2H_4O$ | ++ | n.a. | ++ | ++ | n.a. |
| Acetone | $C_3H_6O$ | ++ | n.a. | ++ | ++ | ++ |
| Acetic acid | $C_2H_4O_2$ | ++ | ++ | n.a. | n.a. | n.a. |
| Ethanol | $C_2H_6O$ | + | n.a. | ++ | ++ | ++ |
| Dimethyl sulfide | $C_2H_6S$ | + | n.a. | + | n.a. | n.a. |
| Isoprene | $C_5H_8$ | + | ++ | n.a. | + | n.a. |
| Monoterpene | $C_{10}H_{16}$ | ++ | ++ | + | + | n.a. |
| 2-Butanone | $C_4H_8O$ | + | n.a. | + | n.a. | n.a. |
| Toluene | $C_7H_8$ | n.a. | ++ | ++ | n.a. | n.a. |
| Formic acid | $CH_2O_2$ | + | n.a. | n.a. | n.a. | ++ |
| Formaldehyde | $CH_2O$ | + | n.a. | n.a. | n.a. | n.a. |

Note: The current understanding of compounds from agricultural management effect is divided into the following levels: ++good (more than 3 papers mentioned), +reasonable (between 1 and 3 papers mentioned), and n.a.: not reported in the literature. References on crop rotation can be found in Sect.3.5 and Sect.3.4 and Sect.2. The mineral fertilization effect can be found in Sect.3.1.1, and the organic fertilization effect in Sect.3.1.2 The tillage effect in Sect.3.2, and the irrigation effect in Sect.3.3.



## 3.1 Fertilization

Using organic or mineral fertilizers affect soil physicochemical properties, microorganism composition (Innerebner et al., 2006; Ros et al., 2006), and microbial activities (Velàzquez-Becerra et al., 2011; Potard et al., 2017). Jiang et al. (2022) reported that bacterial and fungal communities respond to soil physical and chemical properties, which further potentially affect emission rates and compositions of soil BVOC (Janvier et al., 2007; Gray and Fierer, 2012). Studies on applying fertilizers to agricultural fields show that both under-fertilization and over-fertilization can decrease specific BVOC emission rates (Gouinguené and Turlings, 2002), while applying a suitable level of fertilizers can increase BVOC emissions and promote new compounds show up from both crop and agricultural soil (Liang et al., 2023). Variations of BVOC emission patterns and groups of compound are different after applying organic and mineral fertilizers.

### 3.1.1 Mineral fertilization

Applying mineral fertilizers (e.g., nitrogen (N), phosphorus (P), and potassium (K)) can increase soil nutrient status and change pH levels, affecting microorganisms and their physiological state (Stotzky et al., 1976) as well as plant development, and indirectly promoting or inhibiting BVOC emissions. Liang et al. (2023) reported that toluene was abundant in soil fertilized with NPK and NPK+sulfur. Slight variations in the nutrient composition may change the chemical composition and the abundance of BVOC released from the soil (Wheatley et al., 1996, 1997), and BVOC emission change is compound-specific after applying N, P, and K, respectively.

With increased application of N fertilizer, a positive response from both monoterpene and isoprene emission was detected in fields (Litvak et al., 1996), which may be due to the leaf photosynthetic capacity being enhanced (Blanch et al., 2007; Ormeño et al., 2009). A study on oilseed rape reported that the emission rates of lipoxygenase pathway volatiles, $\alpha$-thujene, and acetic acid increased significantly with more nitrogen availability (Veromann et al., 2013). In addition, applying N fertilizer during different phenological periods also showed different BVOC emission responses. During the floral stage of a winter oilseed rape, the originally dominating acetic acid and $\alpha$-pinene are also the compounds whose emissions increased most with an increasing nitrogen level, while adding nitrogen fertilizer during the flowering stage, increased emissions of 3-carene and limonene the most (Veromann et al., 2013). However, all the compounds mentioned above were measured during a pest infestation, and BVOC emission from healthy crops under N fertilization may behave differently. Different to N fertilizers, Fares et al. (2008) mentioned that isoprene emission from aquatic plants decreased when adding P fertilization to the soil.

### 3.1.2 Organic fertilization

Applying organic fertilizers to the soil (e.g., manure, green waste compost, sewage sludge, slurry) improves soil fertility and soil properties, promotes a high bacterial and fungal activity (Asensio et al., 2007; Raza et al., 2017), and affect BVOC emission (Seewald et al., 2010). Organic fertilizers can result in more BVOCs being used as nutrition to support microbial activities and cause a reduction in the total BVOC emissions (Abis et al., 2020). New microbiomes may be imported into the soil while using organic fertilizers. The amount of imported new microbiomes depends on the type of organic fertilizers, and in most





cases, they do not survive long (Sadet-Bourgeteau et al., 2018). Similarly, to organic waste compost has a similar microbial community structure as unfertilized soil, which has a weaker influence than that of other organic fertilizers (Crecchio et al., 2001; Pérez-Piqueres et al., 2006; Ros et al., 2006).

Using slurry and manure as fertilizers changes also soil pH conditions and therefore affects soil bacterial community structures (Case et al., 2018; Nightingale et al., 2022). The emission of dominant compounds, including methanol, acetone, 2-pentanone, and dimethyl sulfide, increases after slurry and manure application (Monard et al., 2020). It has been noted that when applying pig slurry, the proportion of the living bacterial community of $\gamma$-Proteobacteria and Firmicutes both increased largely, which may work as methanotrophs and link to the increase of methanol emissions (Potard et al., 2017). For manure input, methanol is the most emitted compound released from the soil after amendment, and different types of manure treat-

ments led to different modifications in soil microbial diversity and structure (Liu et al., 2007), e.g., volatile sulfur compounds were reported to increase after applying cattle manure (Woodbury et al., 2016). Soil bacteria generally has short-time effect and promote BVOC emission. The emission remains relatively high for one day after manure treatment and seven days after slurry treatment (Potard et al., 2017).

Sewage sludge is easily degraded under aerobic conditions (Wilson et al., 1994), releases nitrogen fast, and may change
microbial diversity (Insam and Merschak, 1997; Six et al., 1999). Applying sewage sludge can result in a high emission of toluene and acetone (Nie et al., 2018; Byliński et al., 2019), and a relatively high level of methanol, ethanol, 2-butanone, and acetaldehyde (Haider et al., 2022). A low level of dimethyl sulfide was detected but with a large degree of difference among studies (Haider et al., 2022).

Burying green waste material into the soil, e.g., vegetable/fruit material, leaf litter, crop residue, etc., can support bacterial
degradation of lignin and cellulose, which can result in high terpene emission (Moreno et al., 2014; Schiavon et al., 2017). Acetone was also detected in soil amended with straw (Zhao et al., 2016).

Soil bacterial communities or the organic status of soils can not be entirely changed by fertilization quickly (Poulsen et al., 2013). Long-term applications had more persistent impacts on soil characteristics (Obriot et al., 2016), plant growth (Clark et al., 2007), and microbial diversity and activity (Francioli et al., 2016; Giacometti et al., 2014). Studies stated that organic

amendment can change soil properties at least after ten years of continued application (Gerzabek et al., 1997; Liu et al., 2013), or after 3 or 4 years of heavy fertilization with organic residues (Tian et al., 2015), and increasing soil bacterial and fungal diversity e.g., Firmicutes, Proteobacteria, and Zygomycota, was observed after more than 20 years of fertilization (Francioli et al., 2016). Therefore, the effect of organic fertilization on BVOC emissions can be a fast change in hours to weeks, and both the chemical composition and abundance of BVOCs change after decades of treatment. Whether BVOC emission increases or

decreases related to long-term fertilization still under debts (Belon et al., 2012; Cambier et al., 2014; Obriot et al., 2016).

## 3.2 Tillage

Tillage can increase soil porosity, decrease soil structure, turn over topsoil layers, and bring crop residues and organic fertilizers into the soil (Unger and Cassel, 1991; Cerdà et al., 2020), which can change the living environment of microbiomes by increasing nutrients and turn over aerobic-anaerobic conditions. Studies suggested that methanol emission may increase after





tillage (Fall and Benson, 1996). Emission of ethanol, butyric acid, acetone, and formic acid increase under anaerobic conditions, while total BVOC released from soil may decrease, resulting from BVOCs produced under aerobic conditions which may serve as an energy source for microorganisms and end up as VOC metabolism (Cleveland and Yavitt, 1998; Owen et al., 2007). It has been shown that more BVOCs are released under anaerobic conditions, e.g., ethanol, butyric acid, acetone, and formic acid (Tang et al., 2019).

## 3.3 Irrigation

Irrigation is a way to meet crop water needs and to help plants resist drought. The impact of irrigation on agricultural BVOC emissions on chemical composition and abundance was little studied. BVOC response paths to re-watering and droughts need to be for different landuses including forests, grasslands, and agricultural plantations.

Generally, the emitting processes of isoprene and some monoterpenes are moisture-dependent (Guenther et al., 2012). Soil 210 may experience different levels of drought without irrigation in non-rainfed agricultural regions. According to the understanding from forest studies, emissions of isoprene and most monoterpenes (e.g., $\alpha$– and $\beta$–pinene) may decline in response to severe drought stress (Brüggemann and Schnitzler, 2002; Lüpke et al., 2017), and for moderate water limitation, an evident change on BVOC emission has not been observed (Loreto and Schnitzler, 2010). However, the emission of specific monoterpene may respond differently to the general pattern, such as the studies in different agricultural plantations reported that the emission 215 of linalool and $\alpha$-phellandrene increase to help plants resist moderate drought stress (Savoi et al., 2016; Palmer-Young et al., 2015; Eirini et al., 2017; Liu et al., 2021b; Wang et al., 2019).

It is worth noticing the positive atmospheric feedback from plant protection mechanisms for fighting drought, e.g., closure of stomata and reduction of green leaf color (Bonn et al., 2019; Flexas and Medrano, 2002), which is reported to enhance atmospheric $O_3$ and $OH$ reactivity, aerosol particles, and cloud properties (Bonn et al., 2019). Maintaining soil water availability at 220 a suitable level through irrigation may slow down the aforementioned atmospheric process.

Irrigation or rainfall after periods of drought will shortly increase the BVOC emissions and recover them to normal levels (Fortunati et al., 2008; Rossabi et al., 2018). Isoprene emission is well matched with biological growth type curves after re-watering soils, while most monoterpenes are related to hydraulic conductivity patterns for stomata opening (Bonn et al., 2019). Over-irrigation or flooding will result in an anaerobic condition, microbial uptake reduction, and emissions of acetaldehyde, 225 ethanol, and acetone were reported to increase from flooded roots and soils (Kreuzwieser et al., 1999). However, their emission may reduce on the ecosystem scale (Faubert et al., 2011). In addition, soil pH was reported to increase after flooding, possibly related to denitrification (Baggesen et al., 2022), which affects the living environment of microbiomes, and further, BVOC emission rates related to microbial activities.

## 3.4 Harvest and post-harvest management

Green leaf volatiles are known to be emitted when plants are damaged as a signal mechanism (Engelberth and Engelberth, 2020). Therefore, these types of BVOCs can be emitted from cut grasslands or from mechanical or chemical destruction of cover crops before planting the following crop. A study on alfalfa observed that a significant increase in emissions of green



leaf volatiles, as well as emission of methanol, acetaldehyde, and acetone, was measured after harvest, and the high level of emission remained for days during residue drying (Warneke et al., 2002).

The typical management of crop residues is to keep them on the fields and let them decay as litter on the soil surface or bury them in topsoil, while tillage provides an organic substrate for microbial activities. Gray et al. (2010) stated that litter decomposition is a source of acetic and formic acids. Previous studies show a large amount of acetone and methanol from crop residues (Jacob et al., 2002; Karl et al., 2003; Gfeller et al., 2013), and the emission of acetone can be even larger if the plant material is heated or wetted or exposed to UV or $O_3$, or if the topsoil is wet (Greenberg et al., 2012; Abis et al., 2021; Warneke

et al., 1999). Formaldehyde is emitted from ground litter under aerobic and anaerobic conditions, related to methanol oxidation during crop decay (Mancuso et al., 2015; Kramshøj et al., 2016). Dimethyl sulfide is another primary compound observed from agricultural lands in field and laboratory studies (Mancuso et al., 2015; Loubet et al., 2022), which may be emitted from aerobic microbial metabolism on litter and soil. Residue burning is another source that quickly releases a considerable amount of BVOC into the atmosphere, dominated by OVOCs (e.g., methanol, formaldehyde) (Ciccioli et al., 2014).

**3.5   BVOC emissions from non-food crops**

Non-food crops are crops not primarily grown for human consumption but serve various other purposes, including cover crops, energy crops, or grasslands. Cover crops are planted a few months between two main crops in the field and are used to maintain soil fertility, limit bare soil periods, and reduce nitrate leaching (Fabrizzi et al., 2005; Kazemeini et al., 2014; Basir et al., 2016). They also affect nitrogen use efficiency and crop production in the following year (Pandiaraj et al., 2015). Most of the time,

cover crops are non-harvested and are left in the field to be killed by freezing or pesticides, and then incorporated into the soil through tillage. Examples include white mustard, oat, common Vetch, or Phacelia. In some cases, temporary grasslands such as alfalfa or clover can be used as cover crops.

BVOC emissions from cover crops are mostly related to their use as green fertilizer and the incorporation of an important amount of fresh biomass into the soil. This has been shown to boost soil microbial activity triggering emissions (e.g. $N_2O$

(Abalos et al., 2022)). Oxygenated Volatile Organic Compounds (OVOCs) are the primary compounds emitted from these cover crops, led by the emission of methanol and acetaldehyde, and followed by acetone (Eller et al., 2011; Bachy et al., 2016;Havermann et al., 2022) (Table 4). Some species can be high isoprene emitters, e.g., water ferns Azolla and velvet beans (Arneth et al., 2008; Silver and Fall, 1995; Monson et al., 1992). Non-food crops have not been extensively studied regarding BVOC emissions except for a few energy crops such as Miscanthus and Switchgrass (Copeland et al., 2012; Eller et al., 2011).



**Table 4.** BVOC emission from non-food crops

| Compounds | Ryegrass | Clover | Miscanthus | Switchgrass | Velvet bean[c] | Azolla[c] |
|---|---|---|---|---|---|---|
| Methanol | 196/66[a] | 320[b] | 3000[b] | 2.49 | | 224.6 |
| Acetaldehyde | 29/9 | | 1000[b] | 0.41 | | |
| Acetone | 19/15 | | 2000[b] | 0.50 | | |
| Acetic acid | 1.3/0 | | 500[b] | | | |
| Isoprene | | | | | 367.8 | 2942 |
| Reference | Havermann et al. (2022) | Custer and Schade (2007) | Copeland et al. (2012) | Eller et al. (2011) | Stoke et al. (1998) | Brilli et al. (2022) |

Note: All units are converted to $\mu$g m$^{-2}$h$^{-1}$. [a]Emissions during heading/ Emission during flowering. [b]Emissions present at the maximum level. [c]The original unit in nmol m$^{-2}$s$^{-1}$ and convert to $\mu$g m$^{-2}$h$^{-1}$. Measurement was conducted on Azolla filiculoides before light treatments.

## 4 Agricultural BVOC modeling approaches and challenges

### 4.1 Numerical modeling of BVOC emissions from agricultural landscapes

BVOC emissions are an essential input that integrates into chemistry and transport models. For example, the Model of Emissions of Gases and Aerosols from Nature (MEGAN) is one of the most widely applied BVOC models set up in climate models, e.g., WRF-Chem, CESM, MOZART, GEOS-Chem (https://bai.ess.uci.edu/megan, last time visit: 03/11/2023).

Most of BVOC emission models are based on an emission factor (EF) approach. The emission factor (EF) is an important input in BVOC models. The EF is a value that describes the abundance of a type of gas/pollutant released to the atmosphere at a specified set of conditions associated with the activity related to the emission of that gas/pollutant (Cheremisinoff, 2011). EFs of agricultural BVOCs are less studied than for tree species. Agricultural BVOC estimation started by Zimmerman (1978) based on general categories of land-use types, e.g., crops, pasture, and orchards. With the availability of a more detailed land cover map and more species-specific measurements, additional crop-specific land use types were included for BVOC estimation, e.g., paddy rice (Lamb et al., 1987). For most crops, e.g., maize, wheat, and rye, a uniform plant type is applied to estimate their BVOC emissions. EF for individual crops is developed and used in the Biogenic Emission Inventory System (Pierce et al., 1998), in which the EF for corn, soybean, wheat, and miscellaneous crops are included, but the emission estimation for crops can be only applied in a few areas where detailed distribution maps for specific crop species are available (Guenther, 2013). A broader category of EF for crops was presented by Karl et al. (2009), where the European major crop species were considered. EFs of OVOCs were applied with a uniform value according to NatAir inventory (Steinbrecher et al., 2000) with more detailed categories for terpenes than other compounds. EF of terpenes for crops in current BVOC models are better developed than those of OVOCs, which may be due to both the greater diversity and more observations that have been reported (Guenther, 2013). MEGAN2.1 has nineteen different BVOC chemical compound classes for 15 categories of plant functional types (including one for cropland) MEGAN3.1 includes additional leaf level trait data for both EF and vegetation parameters (Details described in Guenther et al., 2017) but crop-specific dataset still is a gap. Compared to MEGAN, another model named 'Lund-Potsdam-Jena





General Ecosystem Simulator' (LPJ-GUESS) involved more parameters for crops and agricultural management practices. This model includes a process-based submodel for leaf level emission, which differs from the MEGAN2.1 (Guenther et al. 2012) algorithms, with a developed section for isoprene and monoterpene emission from vegetation (Arneth et al., 2007; Schurgers et al., 2009; Vella et al., 2023). EF of isoprene and monoterpene for crop species of wheat, maize, oilseed rape, sugar beet, etc., are considered, and the values are set according to Karl et al. (2009). In addition, litter effect on soil carbon and nitrogen content, as well as photosynthesis with soil nitrogen limitation, were also included in the model.

To our knowledge both the MEGAN and LPJ-GUESS lack the variation of emission factors with flowering stage and ripening, or management practices, and both models focus on terpenes rather than other compound groups, e.g., OVOCs, which are significantly emitted from crops. Due to the scarce observation from fields and laboratories, especially for species-specific measurements, there remains a large degree of uncertainty regarding the performance of both models on agricultural BVOC estimations for both plants and soils. Further technical details of MEGAN and LPJ-GUESS can be found in Guenther et al. (2012) or Wang et al. (2022), and Hantson et al. (2017), respectively.

## 4.2 Agricultural BVOC estimation challenges

### 4.2.1 Challenges in setting emission factors

The EF for BVOCs is ecosystem/plant species dependent (Guenther et al., 1995, 2012). The standard conditions for EF are, in general, set as $30°C$ for leaf temperature and 1000 $\mu mol m^{-2} s^{-1}$ for photosynthetic photon flux density (PPFD) according to (Guenther et al., 1995), while that for soil BVOC is set as $30°C$ for soil temperature and 6% for moisture level based on Greenberg et al. (2012). The standard conditions were selected due to the key factors controlling plant and microbial production, respectively (Guenther et al., 1995; Tang et al., 2019). EFs for crop BVOC are currently applying the same standard conditions as that used for tree species (Guenther et al., 2012), and lots of studies on crop EFs were calculated by inversing the MEGAN model on an ecosystem or an individual plant scale (Bachy et al., 2016; Gonzaga-Gomez et al., 2019; Liu et al., 2021a). We summarized the crop EFs from published articles and presented the values in Table 5.





**Table 5.** Emission factors for crop BVOCs ($\mu$g m$^{-2}$ h$^{-1}$)

| Compounds | Maize | Wheat | Oilseed rape | Ryegrass |
|---|---|---|---|---|
| Methanol | 25–231 | 162–914 | 262–3557 | 1262 |
| Acetaldehyde | 7 –37 | 18–22 | 4.7–130 | 473 |
| Acetone | 46–57 | 18–85 | 6–266 | 219 |
| Acetic acid | 8 | 6.3 | 0.2–983 | n.a. |
| Ethanol | n.a. | 91 | 25–526 | 101 |
| Dimethyl sulfide | 14.22 | n.a. | n.a. | n.a. |
| Isoprene | 8 | 3.5 | 3.4–272 | 59 |
| Monoterpene | 4 | 6.6 | 3.7–13.7 | n.a. |

Sources of EFs: Maize: Bachy et al. (2016); Wiß et al. (2017); Havermann et al. (2022); Wheat: Gonzaga-Gomez et al. (2021); Loubet et al. (2022); Bachy et al. (2020); Oilseed rape: Havermann et al. (2022), and estimation according to (ICOS) measurements in 2017 (Buysse et al., 2024); Ryegrass: Havermann et al. (2022). All data is presented as the mean value of each study, and the range of EFs is presented as a minimumˇmaximum value among studies. 'n.a.' refers to no data available.

EF for sesquiterpenes have not been presented in Table 5 due to the limit of relative studies, which is a challenge for choosing/estimating high confidence EFs. Sesquiterpenes have not been a focus in previous measurements, possibly related to the quick reactivity and analytical measurement challenges, especially for areas with high oxidant ($O_3$) levels. Bouvier-Brown et al. (2009) and Ciccioli et al. (1999) report that the branch enclosure measurement detected much more emissions of sesquiterpene than the observations at the above plant canopy height. Fares et al. (2012) stated that a better and more quantitative way to

detect sesquiterpene is using leaf/branch enclosure methods from the controlled greenhouse experiment. Therefore, the current agricultural BVOC measurements that acquire data by micrometeorological methods at canopy height may underestimate sesquiterpene emissions. The micrometeorological measurement system may not easily catch high reactivity compounds, but it will not disturb ecosystems during observation, and it observes environmental data (e.g., water, carbon, and energy) and BVOCs simultaneously. Enclosure measurements may be a good way to detect compounds with very high reactivity levels,

but it is difficult to access all parts of a mature plant and keep the original leaf orientation, so BVOC (e.g., light dependent compounds) emissions may be disturbed during measurement (Guenther, 2013).

    The short measurement period is also a challenge to setting reasonable EFs for agricultural ecosystems. Herbaceous crops are fast-growth plants compared to woody plants, therefore emissions among phenology stages can be largely different, and require long-term BVOC measurement coupled with multiple meteorological observations and laboratory EF experimentation

at standard conditions to acquire a better EF with a broader representation of major crop species.

    Parameterization of EFs for agricultural management practices is another challenge. According to previous studies, the EF for different management practices is defined as a value that quantifies the relationship between a chemical compound concentration and the airflow emitted, which is then normalized with respect to one or more reference parameters (e.g. mass of waste to be treated, the emitting surface area, or time) (USEPA, 1995). Examples of EF for different fertilizaters are presented

in Table 6, in which units are in weight of gas per weight of fertilizer applied. To set compound-specific EF based on mechanism



process and build up a connection with possible activity factors $\gamma_{m,i}$, the long-time field measurements before and after each management practice applied, as well as controlling experiments in the lab can provide precise data.

**Table 6.** Examples of VOC emission factors of organic fertilizers

| References | Type | Emission Factor | Unit |
|---|---|---|---|
| SCAQMD (1996) | Green waste | 0.85 | $g\ kg^{-1}$ |
| Büyüksönmez (2012) | Food waste | 2.2 | $g\ kg^{-1}(dryweight)$ |
| Büyüksönmez (2012) | Green waste | 1.41 | $g\ kg^{-1}(dryweight)$ |
| Khosravi et al. (2022) | Green waste | $0.017^*$ | $g\ kg^{-1}$ |
| CIWMB (2007) | Food waste and Green waste | 0.43 to $0.98^*$ | $g\ kg^{-1}$ |
| González et al. (2020) | Sewage Sludge | 6.2 | $kg(C-VOC)\ Mg^{-1}$ |

Note: $^*$ converted from the unit in $lbs\ ton^{-1}$.

### 4.2.2 Uncertainties on meteorological/agricultural inputs for large scale modeling

Meteorological and vegetation data are important inputs in BVOC models, and their spatial and temporal resolution/accuracy largely affect the estimation result. Satellite and reanalysis data with a median or a relatively high resolution are generally applied in local, regional, and global scale modeling, e.g., leaf area index (LAI) from MODIS or Sentinel-3 with the highest resolutions around 500 m and 300 m, respectively. According to Iiames et al. (2015), satellite LAI data had 3% to 20% uncertainties that varied with the vegetation densities. For crop/grass, the LAI uncertainty is around 10% to 15% (Verger et al., 2009), with a misclassification rate of up to 30% among grass and cereal crops (Fang et al., 2013). The rate can be higher between crop and woody vegetation (Fang et al., 2019), which can significantly trigger an underestimation of BVOCs due to species dependent settings. Applying meteorological data acquired from satellite or reanalysis dataset also have uncertainties, but the differences can be smaller than LAI due to the spatial correlation of meteorological parameters.

Agricultural management practices are conducted periodically. Management practices can be identified and characterized from satellite images or with regional documents/reports but that may not be suitable for large scale modeling. Satellite identification depends on the availability of good quality images during the specific period of management practices conducted (Dodin et al., 2021). Data from regional/national annual reports can provide detailed information, e.g., fertilizer types, amounts, and the time of management practice approached, but because of personal data protection, this type of data may not contain detailed spatial information.

## 5 Conceptual model for estimating agricultural BVOC emissions

In section 2.2 and 3, the primary sources, emission processes, impact factors, and current challenges for modeling agricultural BVOC emissions are described. In this section, we present a conceptual model for estimating agricultural BVOC emissions with the effects of management practices (Fig.2).





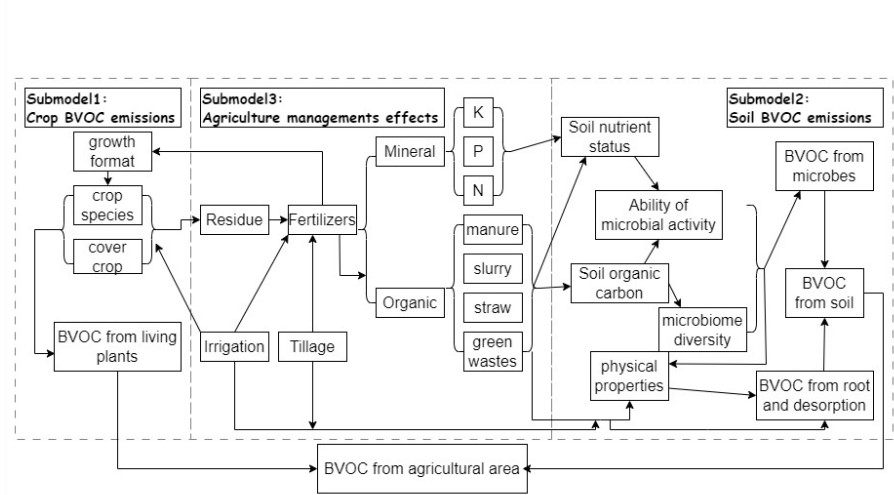

**Figure 2.** Schematic drawing of agricultural BVOC model (Black arrows refer to A effect on B).

This model consists of three submodels to estimate BVOC emissions from plants and soil, respectively, and account for the impacts of agricultural management practices. Agricultural BVOCs are abundant, so we only focused on compounds and compound groups listed in Table 5. Before describing the function and parameters for each submodel in detail, we would like to list the assumptions of this concept model first. 1) Natural factor effects, including air/soil temperature, moisture, light, precipitation, original soil properties before management, atmospheric $CO_2$ and oxidants levels, etc., are as described in Guenther et al. (2012) and Tang et al. (2019). 2) EFs are dependent on compounds and crop species that do not match well with current BVOC models. 3) Agricultural management practices substantially affect soil BVOC emissions. 4) Fertilization and irrigation can be established during any phenology stages.

### 5.1 Crop BVOC emission model

MEGAN was applied as a base model (Eq. 5.1) to estimate BVOC emissions from agricultural herbaceous crops and cover crops. To better match emission patterns, we modified EFs to crop/grass species-dependent values and (Béland and Baldocchi, 2021; Hirooka et al., 2018), which trigger the heat and light distribution among canopy layers. The leaf age is an emission activity factor modified according to crop growth from model simulation.

$$\text{F}_{s,i} = EF_{s,i} \times C_{CE} \times LAI \times \gamma_{s,p,i} \times \gamma_{s,T,i} \times \gamma_{s,SM,i} \times \gamma_{s,CO_2,i} \times \gamma_{s,age,i} \qquad 5.1$$

The emission $F_{s,i}$ of BVOC $i$ from plant species $s$ is presented in Eq. 5.1, where $EF_{s,i}$ is the emission factor of compound $i$, $C_{CE}$ is the canopy environment coefficient, $LAI$ stands for one-sided leaf area index. The series parameters $\gamma_{s,i}$ are activity factors that account for emission responses due to various environmental conditions, including light ($\gamma_{s,p,i}$), leaf temperature ($\gamma_{s,T,i}$), soil moisture ($\gamma_{s,SM,i}$), $CO_2$ inhibition ($\gamma_{s,CO_2,i}$), and leaf age effect ($\gamma_{s,age,i}$).



## 5.2 Soil BVOC emission model

Because microbial decomposition and root emission are two major paths for soil BVOC emission in natural environments, we consider the above two processes as sources in the soil model. Soil organic carbon provides a source for microbial activities, and soil environmental factors, including soil temperature, soil water content, pH, and soil properties, affect decomposition rates (Wieder et al., 2015). In the submodel section for soil BVOC estimation, we follow the model from Tang et al. (2019) to calculate BVOC emissions from agricultural soil ($F_{soil}$) through microbial decomposition and root emission in a condition of natural environments (Eq.5.2).

$$F_{soil} = k_{soil,i} \times f_{soil,i}(WFPS) \times f_{soil,i}(T_s) \times C_{soil,i} - U_{micro,i} \qquad 5.2$$
$$U_{micro,i} = k_u \times f_u(WFPS) \times f_u(T_s) \times C_u$$

Here $k_{soil,i}$ is the relative production rate for BVOC $i$ from the soil, which includes both soil and root emissions. The parameter of $f_{soil,i}(WFPS)$ and $f_{soil,i}(T_s)$ is the soil emission rate affected by soil water content ($WFPS$) and soil temperature ($T_s$), respectively. $C_{soil,i}$ represents the soil carbon pool size. $U_{micro,i}$ represents the BVOC flux due to microbial uptake, which is related to the relative uptake rate ($k_u$), soil uptake rate response to soil temperature ($f_u(T_s)$) and soil water content ($f_u(WFPS)$), and carbon content ($C_u$).

## 5.3 Management practices emission model

Direct measurements presenting the process from applying fertilizer with impacts on soil properties and subsequent changes in BVOC emission have not been reported so far to our knowledge. We can set up a simplified module (gray box) according to studies on soil BVOC source partitioning, as well as the studies on the impacts of agricultural management on the soil environment in the lab (e.g. Abis et al., 2020, 2021; Haider et al., 2022). The BVOC emission induced by different management practices are presented as $F_{m,i}$ in Eq.5.3, which relates to emission factor ($E_{m,i}$, unit in $\mu g\ m^{-2}h^{-1}$), scale factor ($C_{m,i}$, unitless), activity factor ($\gamma_{m,i}$, unitless) and storage VOC during releasing process ($D_{m,i}$, unit in $\mu g\ m^{-2}h^{-1}$). $E_{m,i}$ is a value that depends on the type of management practice, and is decided by the amount of inputs (e.g., N, P, K, water, C) to the agricultural soil. $E_{m,i}$ is a dynamic value according to previous studies mentioned in Sect.3.2, which will change with time passed after management practices are applied, i.e., seven days, a month, and long-term after application. $C_{m,i}$ is the value for correcting $E_{m,i}$ as time passes. $\gamma_{m,i}$ presents the possible factors affecting nutrients and how effectively crops utilize the nutrient inputs, which relate to soil temperature, soil moisture, pH, and nitrogen levels. $D_{m,i}$ describes how much VOC (related to management practices) is taken up by microbiomes in soil and plant storage. $D_{m,i}$ follows ammonia emission when m stands for fertilizer application or organic matter into the soil due to other management practices (Massad et al., 2010). The $D_{max,i}$ below refer to the emission of VOC $i$ reach maximum after management practice $m$, and the parameter t is the time in days, and $\tau$ is the decay time and is set to 2.88 days constantly.

$$F_{m,i} = E_{m,i}C_{m,i}\gamma_{m,i}(t) - D_{m,i} \qquad 5.3$$
$$D_{m,i} = D_{max,i}e^{\frac{-t}{\tau}}$$



Eq.5.3 is an empirical function rather than a mechanistic one due to a lack of evidence for each process. The important path affect $_{m,i}(t)$, e.g., the one between management practice and microbial activity, is not presented. Besides, the crop species applied for rotation can result in different soil nutrient levels and indirectly impact BVOC released from the soil layer (Hirzel et al., 2021), but this effect is not included in the model considering the limited crop combinations that have been studied.

## 6 Conclusions

Agricultural land covers over one-third of terrestrial areas globally, but BVOC emissions from agricultural ecosystems and their feedback to the atmosphere have been little studied. More and more field and laboratory measurements report the important role of agricultural ecosystems on the emission of oxygenated BVOCs, which are tightly related to the air quality and climate, but with less understanding of variations induced by environmental conditions. In this review paper, we summarize the current understanding and datasets regarding agricultural BVOC emission processes. We present a table of emissions during different phenological stages and under various management practices for oilseed rape, wheat, maize, and cover crops. Additionally, we provide a list of crop-specific emission factors for dominant BVOCs. Furthermore, we discuss the current challenges in agricultural BVOC modeling and propose a conceptual model for estimating agricultural BVOC emissions from crops, soil, and different agricultural management practices.

Long-term BVOC measurements or phenology-focused measurements, especially when observed at different height layers, can provide quantitative data to unveil BVOC emission characteristics from both soil and crops during plant development. Laboratory studies can further expand our understanding of BVOC emission processes under various agricultural management effects. This includes investigating responses to factors such as soil pH, nitrogen and carbon content, microbial activities under different management practices, and BVOC emission changes in response to varying soil conditions. Measurements for specific management practices and for different organic matter applications should be conducted, as well as how those vary with time since there is no evidence of their importance in atmospheric chemistry mechanisms.

*Data availability.* Data from publications used in this work are available from the authors upon request ($yang.liu@inrae.fr$ and $raia-silvia.massad@inrae.fr$)

**Appendix A**



**Table A1.** Summary of management practices, their agricultural use, the impacted soil-plant variables in relation to BVOC emissions as well as the processes involved.

| Management Practice | Agricultural Use | Affected soil-plant variables | Processes involved |
|---|---|---|---|
| Species choice in rotations | · Diversification in crop rotations<br>· Economic profitability for farmers | · Biological characteristics of plant | · Plant phenological stages and plant base emission ($D^4$) |
| Cover crops and grass rotations | · Return nutrients to soil<br>· Interrupt pest/disease cycles<br>· Preserve moisture in deep soil layers | · Soil nutrients<br>· Soil water content | · Plant phenological stages and plant base emission ($D^4$) |
| Fertilization | · Meet nutritional needs of crops<br>· Enhance soil carbon storage<br>· Recycle animal (and human) waste | · Soil microbiological characteristics<br>· Soil physico-chemical characteristics [1]<br>· Plant growth | · Soil microbiological emission ($D^4$)<br>· Soil water retention capacity, soil oxic vs anoxic conditions, etc. ($ID^5$)<br>· Plant phenological stages and plant base emission ($D^4$) |
| Irrigation | · Meet plants water demands | · Plant growth<br>· Soil water content | · Plant phenological stages and plant base emission ($D^4$)<br>· Soil emissions via water availability for microbial activity ($ID^5$) |
| Tillage | · Control weeds and pests<br>· Prepare soil for seeding<br>· Distribution of SOC[2] and external organic carbon[3] | · Soil microbiological characteristics<br>· Soil physico-chemical characteristics [1] | · Soil emission via soil physico-chemical characteristics ($ID^5$) |

Note: [1]Soil physico-chemical characteristics includes: soil porosity, pH, soil organic carbon content; [2]SOC: soil organic carbon; [3]External organic carbon includes residues, weeds, etc; [4]D: Direct effect; [5]ID: Indirect effect.

*Author contributions.* Conceptualization: YL, RSM, RC. Writing-original draft: YL. Data preparation: YL, RSM, RC, F.Lafouge, PB.
Writing-review and editing: RSM, RC, LA, BL, AG, BH, CA, F.Levavasseur, AV.

*Competing interests.* The contact author has declared that none of the authors has any competing interests.



*Acknowledgements.* The authors acknowledge the support of the (Agence de l'Environnement et de la Maîtrise de l'Energie (ADEME)), under grant CORTEA 2019 – 1962C0017 (project COVEA) as well as the French PRIMEQUAL program (Programme de recherche in-
terorganisme pour une meilleure qualité de l'air à l'échelle locale), by the French Environment and Energy Management Agency (ADEME), French National Research Agency ANR-21-CE01-0019-01, and by the French Ministry for an Ecological and Solidary Transition under grant 2003C0028 (project RECAPS). And the European Joint programme EJP SOIL funded from the EU Horizon 2020 research and innovation programme (Grant agreement N° 862695) as part of the subproject EOM4SOIL.



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
