# Peer review of "Emissions of biogenic volatile organic compounds from agricultural lands and the impact of land-use and other management practices: A review"

_EGUsphere, 2024_

## Author Comment (AC1)

**We sincerely thank the reviewers for their valuable feedback, and we are especially grateful to the editor for his patience and efforts on our review paper. We have responded to the reviewers' comments point by point and outlined our modification plan to address the corresponding issues as requested.**

Responses to Reviewer 2

*This paper has a highly worthwhile aim with closing an important gap in the literature by reviewing agricultural VOC emissions. But I unfortunately have to agree with the other referee that the manuscript left me unsure how the overview givencan be useful for modeling and the atmospheric chemistry community in general.*

*My main concerns are:*

*- The scope of the review is much narrower than the title suggests. For example, it seems to be focusing purely on field crops that are common in temperate climates. Important BVOC emitters such as fruit trees, or others that are relevant in warmer climates, are ignored.*

**Reply: Thanks for this comment. We have already stated our main aim in line 6 of the abstract in the original text. In the new revision, we will narrow our title to make the aim of the paper clearer.**

*- I think the authors could have done a more thorough job in reviewing the literature and in supporting some of their statements with references.*

**Reply: Thank you for the comment. We would like to affirm that we conducted a detailed review of the topic we focused on. Regarding the comments about the lack of support for some statements, we will carefully go through the paper again and check them in detail.**

*- I am missing a clear direction statement/recommendation of what the numbers given can or should be used/useful for.*

**Reply: We will add the corresponding statements explaining the reasons for showing data as values at the beginning of each section in the revised version. To briefly clarify:**
**The aim of this paper largely dictates the way we present the data. Our goal is to summarize the current processes related to agricultural arable crops, soil, and management practices, highlighting which research areas remain unclear and require further study. For modeling purposes, we realized that using a single PFT definition (e.g., MEGAN v2.1) and less focus on OVOC (Karl et al., 2009) do not meet the needs or match real environmental situations. Therefore, we present the current knowledge on emission factors for arable crops in Table 5 of our review paper. Additionally, the emission values in Tables 1 and 2 clarify our statement on the necessity for species-specific emission factors for each compound. As mentioned before, we can improve the logical presentation of data in each section in the revised version.**

*- In conclusion, I am afraid that I cannot recommend the manuscript's publication in its current form. To give the authors more time to revise I would rather reject the manuscript.*

*Some concrete points:*

*l. 14 Guo et al. is a modeling study. You could add a reference of actual measurements showing the relevance of BVOCs for urban air quality. But I am not sure if urban biogenics are the best introduction into this paper on agricultural emissions.*

**Reply: Thanks. We will consider this comment in the revised version.**

*l. 15 citations are missing for the examples given*

**Reply: The reference for line 15 was given in line 754.**

*l. 18-20 for the impact of agricultural BVOCs on atmospheric chemistry, you could also cite*

*Bsaibes, S., Gros, V., Truong, F., Boissard, C., Baisnée, D., Sarda-Esteve, R., Zannoni, N., Lafouge, F., Ciuraru, R., Buysse, P., Kammer, J., Gomez, L. G., and Loubet, B.: Characterization of Total OH Reactivity in a Rapeseed Field: Results from the COV3ER Experiment in April 2017, Atmosphere, 11, 261, https://doi.org/10.3390/atmos11030261, 2020.*

*Pfannerstill, E. Y., Arata, C., Zhu, Q., Schulze, B. C., Woods, R., Seinfeld, J. H., Bucholtz, A., Cohen, R. C., and Goldstein, A. H.: Volatile organic compound fluxes in the agricultural San Joaquin Valley – spatial distribution, source attribution, and inventory comparison, Atmos. Chem. Phys., 23, 12753–12780, https://doi.org/10.5194/acp-23-12753-2023, 2023.*

**Reply: We appreciate and agree with your suggestions.**

*l. 30 maybe another relevant paper to cite here: Rinnan, R. and Albers, C. N.: Soil Uptake of Volatile Organic Compounds: Ubiquitous and Underestimated?, J. Geophys. Res. Biogeosci., 125, e2020JG005773, https://doi.org/10.1029/2020JG005773, 2020.*

**Reply: We will add this in the corresponding section.**

*l. 42: Please also clarify that you exclude emissions from animal agriculture like dairies and openly stored silage, which are a big source of VOCs in some regions, and address the above mentioned scope comment.*

**Reply: Thank you for this comment. We will add this to make it more clear to the audience.**

*Table 1: In some regions of the world, citrus and other fruit trees are important for agriculture. They emit large amounts of highly reactive monoterpenes and should therefore be included in this review, I think. Otherwise, the review should specify already in the title and make clear that this is just for "field crops" and excludes fruits and other tree crops.*

*On citrus monoterpenes: e.g.*

*Fares, S., Gentner, D. R., Park, J.-H., Ormeno, E., Karlik, J., and Goldstein, A. H.: Biogenic emissions from Citrus species in California, ATMOSPHERIC ENVIRONMENT, 45, 4557–4568, https://doi.org/10.1016/j.atmosenv.2011.05.066, 2011;*

*Gentner, D. R., Ormeño, E., Fares, S., Ford, T. B., Weber, R., Park, J.-H., Brioude, J., Angevine, W. M., Karlik, J. F., and Goldstein, A. H.: Emissions of terpenoids, benzenoids, and other biogenic gas-phase organic compounds from agricultural crops and their potential implications for air quality, Atmos. Chem. Phys., 14, 5393–5413, https://doi.org/10.5194/acp-14-5393-2014, 2014. [This paper's SI has a big table with emission factors for many agricultural BVOC emissions]*

**Reply: As stated in line 6 of the abstract, arable crops are the key species we focus on in this paper. For citrus, multiple papers, including the two mentioned by the reviewer, clearly describe their significant contribution. Interested readers can refer to these articles for more details.**

*Table 2, footnote a: Since there is no publication cited here, the data need to be published somewhere with a doi as specified in the data availability policy of ACP.*

**Reply: Since the references are too long to show, we will add a table in the supplement to make the usage clearer.**

*Table 2: It is not clear to me why forest soil emissions are included here (if you wanted to include data from forest soils and not just agricultural soils, there are far more studies of forest soil VOC emissions from many different places that would need to be included). Also, I am not sure if the few studies included in this table are all there is on VOC emissions from agricultural soils. Just a 2-minute internet search gave me more publications that report emission rates from agricultural soils, e.g.*

*Juan Zhao, Zhe Wang, Ting Wu, Xinming Wang, Wanhong Dai, Yujie Zhang, Ran Wang, Yonggan Zhang, Chengfei Shi, Volatile organic compound emissions from straw-amended agricultural soils and their relations to bacterial communities: A laboratory study, Journal of Environmental Sciences, Volume 45, 2016, Pages 257-269, https://doi.org/10.1016/j.jes.2015.12.036.*

*And the Abis et al. 2020 paper that the authors cite in other contexts. If there are reasons to exclude such studies from the compilation, it would be helpful if the authors defined the criteria of their selection of data more clearly.*

**Reply: Section 2.1 focuses on bare soil as the title described, which data presented are without the effects of management practices. The papers mentioned by Reviewer 2 focus on fertilization mixed with soil, which is not what we intend to present here. We will cite Zhao's article in the section on organic fertilization, where Abis et al., 2020 is also cited.**

*2.3 flowering emissions have also been discussed in some publications on citrus*

*l.95: "emission rates from this period" – do you mean "soil emission rates"?*

*l. 175: The sentence is grammatically incorrect and content-wise unclear.*

*Tables 4/5: This paper's SI has a big table with emission factors for many crop BVOC emissions that should not be ignored in this review: Gentner, D. R., Ormeño, E., Fares, S., Ford, T. B., Weber, R., Park, J.-H., Brioude, J., Angevine, W. M., Karlik, J. F., and Goldstein, A. H.: Emissions of terpenoids, benzenoids, and other biogenic gas-phase organic compounds from agricultural crops and their potential implications for air quality, Atmos. Chem. Phys., 14, 5393–5413, https://doi.org/10.5194/acp-14-5393-2014, 2014.*

*Fig. 1 I like the sentiment, but the realization could be improved... where do H2O and CO2 go, for example? it is a bit much information in a small space, making it confusing. Do the animals shown contribute anything? I don't see animal agriculture discussed in the paper.*

*Fig. 2: I would refrain from using Comic Sans in a scientific figure.*

**Reply: We will revise section 2.3 accordingly.**

*- Data availability: upon request is not acceptable according to the data policy of ACP. Please make the data available in a publicly accessible repository.*

**Reply: We understand the importance of making data readily available and are committed to transparency. While we can publish the data from our own research, some of the data included in this review are sourced from other authors and groups. To respect their intellectual property, we indicated that these data are 'available upon request.' We will clarify this in the manuscript and will gladly assist in facilitating access to the data through the corresponding groups, should any requests arise.**

---

## Author Comment (AC2)

**We sincerely thank the two reviewers for their valuable feedback, and we are especially grateful to the editor for his patience and efforts on our review paper. We have responded to the reviewers' comments point by point and outlined our modification plan to address the corresponding issues as requested.**

Responses to Reviewer 1

*This paper addresses a very interesting topic, promising a review of the long neglected field of agricultural VOC emissions. The best thing about this paper is that it present many citations, and highlights some of the complex issues associated with modeling such BVOC emissions. Unfortunately, I found the paper to be too qualitative and confusing to be considered a review, and believe that the changes needed to improve the paper are more than those possible with "major review". I am afraid that I cannot recommend publication.*

*Major comments:*

*The paper is frequently confusing in what is presented, and the numbers presented are often not defined or useful. Some examples:*

● *Table 1 presents "BVOC emissions", with units μg/m2(leaf)/h, but the units are for emission factors (EFs), not emissions. More seriously, the numbers are just numbers. There is no information on the environmental conditions under which these EFs were measured. I am guessing that these are not emission potentials (EPs) within either the earlier Guenther systems (ie at 30 deg. C, full sunlight) or the newer MEGAN EPs, so how can we use these numbers? What are they for?*

**Reply: Table 1 presents emissions rather than emission factors, as mentioned in the table caption. The conversion principle of the emission unit is explained in a footnote under the table. The emission unit can be μg/m²(leaf)/h, as seen in Pihlatie et al., 2005, and Zhu et al., 2022.**

Pihlatie, M., Ambus, P., Rinne, J., Pilegaard, K., and Vesala, T. (2005). Plant-mediated nitrous oxide emissions from beech (Fagus sylvatica) leaves. *New Phytol*. 168, 93–98. doi: 10.1111/j.1469-8137.2005.01542.x

Zhu C.F., Luo HD , Luo LC , Wang KY, Liao Y , Zhang S, Huang SS, Guo XM, Zhang L. Nitrogen and Biochar Addition Affected Plant Traits and Nitrous Oxide Emission From Cinnamomum camphora. *Frontiers in Plant Science*,13, 2022. 10.3389/fpls.2022.905537

*Table 1 also has negative emissions for some species/periods. What are these? My first guess would be deposition, but then why aren't the species deposited at other stages? In any case, nothing is explained.*

**Reply: We appreciate your insight regarding the negative emissions. These reflect depositions, and we will provide a detailed explanation in Section 2 to clarify this point. Thank you for highlighting this area that could benefit from further clarification.**

● *Tables 2, 4. Same points as with Table 1.*

**Reply: We understand that the distinction between emissions and emission factors might have been unclear. Our intention was to present emissions in Tables 2 and 4, as we aimed to highlight the significance of species-specific emission factors for arable crops, bare soil, and cover crops. We will ensure this distinction is made clearer in the revised manuscript.**

● *Table 5. Are these emission factors for 30C, 1000 μmole/m2/h, or for MEGAN2 conditions, or something else?*

**Reply: The details that Reviewer 1 mentioned had already been declaimed in L300-301.**

● *In Sect. 4.2.1, L296, it is stated that the standard conditions for EFs are 30C and 1000 μmole/m2/s, but MEGAN2 uses a much more complicated definition. I assume that Table 5 is for 30C, but as with other Tables, this is not explicit.*

**Reply: 1. We can revise this sentence for clarity to avoid any potential misunderstanding. According to the definition of the standard condition for emission factors (EF) in MEGAN 2, as detailed on Page 3181 of Guenther et al., 2006, we explained in Lines 300 and 301 of our paper how we derived EFs for crop species when data were not obtained from a meta-analysis.**
**2. Yes, this is correct. All tables, except for Table 5, present emissions rather than emission factors. These emissions are not intended to be directly inputted into MEGAN or other models.**

● *Sect 3, L131 "Studies show...."? This important section makes statements about BVOC emissions, but no citations are given. Which studies? Table 3 is referred to, but no citations appear there. (In the footnote to this table there is further forwarding to different sections later in the text, which makes the table awkward to read. It would be better with a Table row giving such information.)*
**Reply: Thanks for this comment. We will add an appendix table to show the references used in this table.**

● *The paper makes very little mention of the differences or issues surrounding leaf-scale versus canopy scale versus ecosystem scale emissions. Thus the sentence starting on L225 suddenly mentions that emissions may be reduced on an ecosystem scale, but no real explanation is given.*
**Reply: We will revise the description to make the information from different scales clearer. Thank you for pointing out Line 225, we will add an explanation for the cited statement**

● *Section 4.1 "Numerical modeling of BVOC" is also confusing. On L266 they define EFs as the "abundance" of a type of of gas/pollutant, but one would normally define EFs in mass released per unit leaf-area or leaf-mass per unit of time. The cited "Cheremisinoff 2011" paper isn't in the reference list, and I would anyway have expected a Guenther-type reference here. On L271 the paper states that "a uniform plant type is applied", but where, by whom? On L272-274 I am not sure what the link is between the Pierce statement and the Guenther 2013 reference.*
**Reply: Thanks. We will modify the points throughout your comments.**

● *The text is very qualitative, e.g. on L71 we read "emitted at relatively low rates", on L89 we have a "considerably higher emission rate". On L146 we read that "toluene is abundant in soil", but are there substantial emissions, also in comparison to e.g. road traffic emissions? Very much of the important section 3 is qualitative, making it difficult to know if emissions are really potentially important, or simply something somebody measured, somewhere. Similarly, on L237 we read that "a large amount of acetone..." is possible, but large compared to what?*
**Reply: Thank you for your detailed observations. We will carefully revise the text to ensure that qualitative statements are more precise and supported by quantitative data where possible.**

● *L279-281 states that MEGAN2.1 has 19 VOC compounds for 15 plant categories; are these 285 EFs supported by measurements? How many are?! I would have hoped that a "review" of such BVOC emissions, and with Alex Guenther as coauthor, would have provided more background to such issues.*
**Reply: 'MEGAN2.1 has 19 VOC from 15 plant categories' mentioned in Guenther et al 2012. The 19*15 = 285 EFs can be found either in the previously mentioned paper or by emailing the Guenther group. In any case, this review paper we submitted focuses solely on agricultural land; we do not provide information for other landscapes.**

*In Section 4.2.2 I missed a discussion of the very real uncertainties associated with the specification of agricultural events: dates of sowing, emergence, growth, and fertilizer application. I know this is mentioned in the last paragraph, but the wording is rather vague. Is there any realistic hope of using satellite data to specify phenology and agricultural practices for European and/or global scale modeling? What would be needed to make progress in this field?*
**Reply: In Section 4.2.2, we discussed the limitations of agricultural management-related data in lines 337 to 344. We did not suggest that 'there is no realistic hope of using satellite or statistical data.' On the contrary, we strongly encourage Reviewer 1 and other readers to consider using these data for larger-scale modeling. However, as a review paper, we also have the responsibility to highlight potential challenges for future use, so that scientists in related fields can address**

**and improve upon them. Considering that other audiences may share the same concerns as Reviewer 1, we can consult with agronomists and incorporate their current solutions into the corresponding section for a more detailed discussion.**

● *Section 5.1 (L357) starts "MEGAN was applied as a base model (Eq. 5.1) to estimate BVOC emissions from agricultural herbaceous crops...", but MEGAN isn't applied here. Further, if I understood right Table 5 gives emission factors using the older 30 deg C definition of emission factors, whereas MEGAN requires much more complex conditions.*
**Reply: 1. MEGAN v2.1 (Guenther et al., 2012) includes a PFT15 function specifically for crops.**
**2. The emission factor (EF) and canopy model for light and radiation distribution will be modified. In lines 358-359, a portion of the sentence was omitted, specifically the phrase 'the canopy model for LAI depth distribution.' As a result, only two citations related to crop-specific growth formats were left in that section.**
**3. We appreciate your perspective regarding the '30°C' statement. In line 301, we aimed to clarify that our approach involves inverting MEGAN, considering all relevant meteorological and environmental inputs. We will revise this section to ensure that the process and reasoning are fully transparent and accurately presented.**

● *Section 5.1 continues (L358) to say "we modified EFs to crop/grass species-dependent values", but no details are given of the resulting EFs. This is all very confusing!*
**Reply: We agree that this part may cause confusion because it should not involve a conceptual model in a review paper rather than a research paper. We will modify the entire Section 5 conceptual model to be more based on published papers.**

● *Section 5.3 was also wordy but vague; which information here can be used, and/or what is needed before we can use such information.*
**Reply: Thank you. We will remove the non-published paper-based information from this review paper as previously mentioned.**

● *Section 6, Conclusions states the paper presents "a table of emissions during different phenological stages", and that they "provide a list of crop-specific emission factors for dominant BVOCs", but as noted above the numbers provided are confusing and probably not useful.*
**Reply: Thank you. We will remove the non-published paper-based information from this review paper as previously mentioned. We will also mention this statement as potential research points for further agricultural VOC measurement approaches.**

● *Data availability: these days data should be provided in SI, or via zenodo I think. Available from the authors on request is always dependent on the availability of the authors.*
**Reply: Thank you for this comment. We did not expect the reviewer to interpret this statement in this way. It is easy to arrange data from a research paper because all data are measured by the authors or their team. However, as a review paper, we have data from other groups, which means we have the right to use it but must respect their work. Therefore, we state here that the data is available from the authors upon request.**

*Smaller points:*
*The English needs a thorough revision. There are many cases where cases don't match (leaf versus leaves for example), and some cases where the sentences don't make sense (e.g. L82-84). Other examples: L86 - I guess you mean emergence and not emergency? ; L109: what does "besides the fate of VOCs" mean here? ; L140: "and promote new compounds show" isn't English; L150: Does "a positive response" mean increased emission rate? ; L166 delete "to" from "to organic waste"*
*Table 1: it would be useful for non-agriculturalists to give the approximate time-periods of the different stages.*
**Reply: We appreciate your observation and agree that this is an important point. We will make the necessary revisions to improve clarity.**

*L96. The paper states that "The soil continues emitting BVOCs during plant growth and ripening, but emission rates from this period have not been reported so far to our knowledge." So, how do you know that BVOCs are still emitted?*

**Reply: We mentioned that soil continues to emit VOCs even when plants are present in the field because: 1. Microbial activity, soil moisture, and particle absorption are primary contributors to soil VOC emissions (e.g., Tang et al., 2019), which depend on soil properties and the amount of soil organic matter. Plants, particularly their residues and root systems, can increase soil organic matter content to some extent.**

**2. Agricultural soil measurements provide insights under specific conditions, such as during a heatwave with wheat in the field, as observed by Schade et al., 2004 (https://doi.org/10.1016/j.atmosenv.2004.08.017). We can add these references in the corresponding section and clearly state the specific conditions and influencing factors.**

*L141. The last sentence is so vague ("Variations of BVOC..") as to be meaningless.*
**Reply: Thanks for the comment. We will revise accordingly.**

*L217-220. This is a bit vague and unclear. If drought reduces BVOC emissions, wouldn't one get less secondary organic aerosol, not more?*
**Reply: In line 213 - 216, we already started to talk about increase effects rather than reduce effects. And the positive to secondary organic aerosol can be find more details from the paper we cited Bonn et al., 2019.**

*L223. The text here and around relies a lot on Bonn et al., 2019, but that paper only dealt with trees. Also, many monoterpene emissions are not under stomatal control, being rather stored in pools within the leaf (e.g. Niinemets et al., 2004, Guenther et al - many papers).*
**Reply: Bonn et al., 2019 was cited in the paragraph discussing how stressed VOCs contribute to particle formation L217-220. We did not cite this paper in the section that discusses the types of VOCs emitted by crops during drought. However, we will clarify this distinction for the readers.**

*L247. The statement "Cover crops are planted a few months between two main crops" is likely true in France, but do all countries have two main crops, with a few months between them? In general, this paper has little consideration of climatological differences between even parts of Europe, let alone the globe.*
**Reply: We will narrow and state the regional information in our title and material section. About cover crop, *'is likely true in France, but do all countries have two main crops'*, see reference: e.g. Fendrich et al., 2023.**
*Arthur Nicolaus Fendrich, Francis Matthews, Elise Van Eynde, Marco Carozzi, Zheyuan Li, Raphael d'Andrimont, Emanuele Lugato, Philippe Martin, Philippe Ciais, Panos Panagos. From regional to parcel scale: A high-resolution map of cover crops across Europe combining satellite data with statistical surveys, Science of The Total Environment, 873, 2023, 162300, https://doi.org/10.1016/j.scitotenv.2023.162300.*
**Although cover crop is applied in short period, and normally during winter time, but as an approach of management practice with less focusing, we believe it worthy and should be mentioned in our review paper.**

*L262: Usually the "and" can be dropped between chemistry and transport models.*
*L263. Give MEGAN a proper reference.*

*L288 states "To our knowledge" about MEGAN and LPJ-GUESS, but you have the lead author of MEGAN on the author list. I am sure Alex Guenther knows. And it would not take much effort to ask the LPJ developers about the EFs being discussed.*

*L323: What is "airflow" emitted?*

**Reply: Thanks for the comment. We will revise accordingly.**

*L371: why are natural environments relevant here. Agricultural land us far from natural.*

*L365 on, Section 5.2. Is WFPS a good indicator of soil moisture? Soils with the same WFPS can have very different soil water pressure values.*

**Reply: Thanks for the comment. We will remove the Section 5 and only focusing the publication review in new version. About the Section 5.2, as we mentioned in line 369, this model '***follow the model from Tang et al. (2019) to calculate BVOC emission from soil***'. We believe they did enough experiments for parameterization.**

*L395-397 - this text is unclear. Re-word.*

*General: why are italics used for words such as "dry weight" in Table 6, and in units throughout the text, e.g., for g, kg, m or h? Also, g and kg and VOC should not be italic.*

**Reply: Thanks for the comment. We will revise accordingly.**

---

## Author Comment (AC3)

**We sincerely thank the two reviewers for their valuable feedback, and we are especially grateful to the editor for his patience and efforts on our review paper. We have responded to the reviewers' comments point by point and outlined our modification plan to address the corresponding issues as requested.**

Responses to Reviewer 2

*This paper has a highly worthwhile aim with closing an important gap in the literature by reviewing agricultural VOC emissions. But I unfortunately have to agree with the other referee that the manuscript left me unsure how the overview givencan be useful for modeling and the atmospheric chemistry community in general.*

*My main concerns are:*

*- The scope of the review is much narrower than the title suggests. For example, it seems to be focusing purely on field crops that are common in temperate climates. Important BVOC emitters such as fruit trees, or others that are relevant in warmer climates, are ignored.*

**Reply:** Thanks for this comment. We have already stated our main aim in line 6 of the abstract in the original text. In the new revision, we will narrow our title to make the aim of the paper clearer. We propose as a new title:  Current Insights into Biogenic VOC Emissions from Arable Crops and the potential impacts of management practices: A Review.

*- I think the authors could have done a more thorough job in reviewing the literature and in supporting some of their statements with references.*

**Reply:** Thank you for the comment. We would like to affirm that we conducted a detailed review of the topic we focused on. Regarding the comments about the lack of support for some statements, we will carefully go through the paper again and check them in detail as also highlighted by some of the replies to reviewer #1.

*- I am missing a clear direction statement/recommendation of what the numbers given can or should be used/useful for.*

**Reply:** As mentioned by reviewer # 1 and our reply to that, we acknowledge that it is not very clear what is the end use of the numbers given. Our objective is to give orders of magnitude for emissions from arable crops and practices, to collect the data available in the literature today concerning this landuse and to suggest some ideas for using them into models.
We will add the corresponding statements explaining the reasons for showing data as values at the beginning of each section in the revised version. To briefly clarify:
The aim of this paper largely dictates the way we present the data. Our goal is to summarize the current processes related to agricultural arable crops, soil, and management practices, highlighting which research areas remain unclear and require further study. For modeling purposes, we realized that using a single PFT definition (e.g., MEGAN v2.1) and less focus on OVOC (Karl et al., 2009) do not meet the needs or match real environmental situations. Therefore, we present the current knowledge on emission factors for arable crops in Table 5 of our review paper. As mentioned before, we can improve the logical presentation of data in each section in the revised version.

*- In conclusion, I am afraid that I cannot recommend the manuscript's publication in its current form. To give the authors more time to revise I would rather reject the manuscript.*

*Some concrete points:*

*l. 14 Guo et al. is a modeling study. You could add a reference of actual measurements showing the relevance of BVOCs for urban air quality. But I am not sure if urban biogenics are the best introduction into this paper on agricultural emissions.*

**Reply:** Thanks. We will consider this comment in the revised version.

*l. 15 citations are missing for the examples given*

**Reply:** The reference for line 15 was given in line 754.

*l. 18-20 for the impact of agricultural BVOCs on atmospheric chemistry, you could also cite*

*Bsaibes, S., Gros, V., Truong, F., Boissard, C., Baisnée, D., Sarda-Esteve, R., Zannoni, N., Lafouge, F., Ciuraru, R., Buysse, P., Kammer, J., Gomez, L. G., and Loubet, B.: Characterization of Total OH Reactivity in a Rapeseed Field: Results from the COV3ER Experiment in April 2017, Atmosphere, 11, 261, https://doi.org/10.3390/atmos11030261, 2020.*

*Pfannerstill, E. Y., Arata, C., Zhu, Q., Schulze, B. C., Woods, R., Seinfeld, J. H., Bucholtz, A., Cohen, R. C., and Goldstein, A. H.: Volatile organic compound fluxes in the agricultural San Joaquin Valley – spatial distribution, source attribution, and inventory comparison, Atmos. Chem. Phys., 23, 12753– 12780, https://doi.org/10.5194/acp-23-12753-2023, 2023.*

**Reply:** We appreciate and agree with your suggestions.

*l. 30 maybe another relevant paper to cite here: Rinnan, R. and Albers, C. N.: Soil Uptake of Volatile Organic Compounds: Ubiquitous and Underestimated?, J. Geophys. Res. Biogeosci., 125, e2020JG005773, https://doi.org/10.1029/2020JG005773, 2020.*

**Reply**: We will add this in the corresponding section.

*l. 42: Please also clarify that you exclude emissions from animal agriculture like dairies and openly stored silage, which are a big source of VOCs in some regions, and address the above mentioned scope comment.*

**Reply:** Thank you for this comment. We will add this to make it more clear to the audience.

*Table 1: In some regions of the world, citrus and other fruit trees are important for agriculture. They emit large amounts of highly reactive monoterpenes and should therefore be included in this review, I think. Otherwise, the review should specify already in the title and make clear that this is just for "field crops" and excludes fruits and other tree crops.*

*On citrus monoterpenes: e.g.*

*Fares, S., Gentner, D. R., Park, J.-H., Ormeno, E., Karlik, J., and Goldstein, A. H.: Biogenic emissions from Citrus species in California, ATMOSPHERIC ENVIRONMENT, 45, 4557–4568, https://doi.org/10.1016/j.atmosenv.2011.05.066, 2011;*

*Gentner, D. R., Ormeño, E., Fares, S., Ford, T. B., Weber, R., Park, J.-H., Brioude, J., Angevine, W. M., Karlik, J. F., and Goldstein, A. H.: Emissions of terpenoids, benzenoids, and other biogenic gas-phase organic compounds from agricultural crops and their potential implications for air quality, Atmos. Chem. Phys., 14, 5393–5413, https://doi.org/10.5194/acp-14-5393-2014, 2014. [This paper's SI has a big table with emission factors for many agricultural BVOC emissions]*

**Reply:** As stated in line 6 of the abstract, arable crops are the key species we focus on in this paper and the title was changed accordingly. For citrus, multiple papers, including the two mentioned by the reviewer, clearly describe their significant contribution. Interested readers can refer to these articles for more details.

*Table 2, footnote a: Since there is no publication cited here, the data need to be published somewhere with a doi as specified in the data availability policy of ACP.*

**Reply** The footnote a in Table 2 was published in Open discussion section of EGUsphere, we can added the reference in the corresponding area:
*Buysse, P., Loubet, B., Ciuraru, R., Lafouge, F., Durand, B., Zurfluh, O., Décuq, C., Fanucci, O., Gonzaga Gomez, L., Gueudet, J.-C., Bsaibes, S., Zannoni, N., and Gros, V.: First measurements of ecosystem-scale biogenic volatile organic compound fluxes over rapeseed reveal more significant terpenoid emissions than expected, EGUsphere [preprint], https://doi.org/10.5194/egusphere-2023-2438, 2024.*

*Table 2: It is not clear to me why forest soil emissions are included here (if you wanted to include data from forest soils and not just agricultural soils, there are far more studies of forest soil VOC emissions from many different places that would need to be included). Also, I am not sure if the few studies included in this table are all there is on VOC emissions from agricultural soils. Just a 2-minute internet search gave me more publications that report emission rates from agricultural soils, e.g.*

*Juan Zhao, Zhe Wang, Ting Wu, Xinming Wang, Wanhong Dai, Yujie Zhang, Ran Wang, Yonggan Zhang, Chengfei Shi, Volatile organic compound emissions from straw-amended agricultural soils and their relations to bacterial communities: A laboratory study, Journal of Environmental Sciences, Volume 45, 2016, Pages 257-269, https://doi.org/10.1016/j.jes.2015.12.036.*

*And the Abis et al. 2020 paper that the authors cite in other contexts. If there are reasons to exclude such studies from the compilation, it would be helpful if the authors defined the criteria of their selection of data more clearly.*

**Reply:** Section 2.1 focuses on bare soil as the title described, which data presented are without the effects of management practices. The papers mentioned by Reviewer 2 focus on fertilization mixed with soil, which is not what we intend to present here. We will cite Zhao's article in the section on organic fertilization, where Abis et al., 2020 is also cited. The reference to forests will be removed to be more consistent as suggested.

*2.3 flowering emissions have also been discussed in some publications on citrus*

**Reply:** As stated above, citrus trees are not included here.

*l.95: "emission rates from this period" – do you mean "soil emission rates"?*

**Reply:** Yes this is what is meant. Will be changed.

*l. 175: The sentence is grammatically incorrect and content-wise unclear.*

**Reply:** This sentence should read:

> Methanol is the predominant compound emitted from soil following manure amendments, with different types of manure treatments causing distinct changes in soil microbial diversity and structure (Liu et al., 2007). For example, the application of cattle manure has been shown to increase volatile sulfur compound emissions (Woodbury et al., 2016).

*Tables 4/5: This paper's SI has a big table with emission factors for many crop BVOC emissions that should not be ignored in this review: Gentner, D. R., Ormeño, E., Fares, S., Ford, T. B., Weber, R., Park, J.-H., Brioude, J., Angevine, W. M., Karlik, J. F., and Goldstein, A. H.: Emissions of terpenoids, benzenoids, and other biogenic gas-phase organic compounds from agricultural crops and their potential implications for air quality, Atmos. Chem. Phys., 14, 5393–5413, https://doi.org/10.5194/acp-14-5393-2014, 2014.*

**Reply:** We will verify this paper and amend accordingly.

*Fig. 1 I like the sentiment, but the realization could be improved... where do H2O and CO2 go, for example? it is a bit much information in a small space, making it confusing. Do the animals shown contribute anything? I don't see animal agriculture discussed in the paper.*

**Reply:** $CO_2$ and $H_2O$ are specific to the chemistry part which is perhaps not the focus of this paper and could be simplified. Animals contribute to manure and slurry production which are then spread on arable crops. Perhaps the cows could be put in a separate part of the figure or greyed.

*Fig. 2: I would refrain from using Comic Sans in a scientific figure.*

**Reply:** Noted and will be changed.

*- Data availability: upon request is not acceptable according to the data policy of ACP. Please make the data available in a publicly accessible repository.*

**Reply:** We understand the importance of making data readily available and are committed to transparency. While we can publish the data from our own research, some of the data included in this review are sourced from other authors and groups. To respect their intellectual property, we indicated that these data are 'available upon request.' We will clarify this in the manuscript and will gladly assist in facilitating access to the data through the corresponding groups, should any requests arise.

---

## Author Comment (AC4)

**We sincerely thank the two reviewers for their valuable feedback, and we are especially grateful to the editor for his patience and efforts on our review paper. We have responded to the reviewers' comments point by point and outlined our modification plan to address the corresponding issues as requested.**

*This paper addresses a very interesting topic, promising a review of the long neglected field of agricultural VOC emissions. The best thing about this paper is that it present many citations, and highlights some of the complex issues associated with modeling such BVOC emissions. Unfortunately, I found the paper to be too qualitative and confusing to be considered a review, and believe that the changes needed to improve the paper are more than those possible with "major review". I am afraid that I cannot recommend publication.*

**Reply:** We understand that the analysis in this paper is not as quantitative as expected. However we do not aim to present solely a quantitative assessment. The objective is rather to review existing data (which we show that is still scarce today for agricultural land uses) and to propose a theoretical approach to estimate BVOC emissions for agricultural land uses. The conclusion being that it is important to account for agricultural practices and to conduct more measurements in this area to have more data.

*Major comments:*

*The paper is frequently confusing in what is presented, and the numbers presented are often not defined or useful. Some examples:*

● *Table 1 presents "BVOC emissions", with units μg/m2(leaf)/h, but the units are for emission factors (EFs), not emissions. More seriously, the numbers are just numbers. There is no information on the environmental conditions under which these EFs were measured. I am guessing that these are not emission potentials (EPs) within either the earlier Guenther systems (ie at 30 deg. C, full sunlight) or the newer MEGAN EPs, so how can we use these numbers? What are they for?*

**Reply:** Table 1 presents emissions rather than emission factors, as mentioned in the table caption. This table is not for modelling purposes but rather to show the different measurements present in the literature today and the orders of magnitude of those emissions for the different crops and the different phenological stages. We could add a specific column in this table specifying briefly the environmental conditions for each experiment.  About the conversion principle of the emission unit is explained in a footnote under the table. For example, VOC is measured in units as $\mu g \ gdw^{-1} \ h^{-1}$, acquiring the value of weight per $m^2$ of leaf from their papers or contacting their corresponding authors to get the corresponding values, then multiplying the values to get the unit we presented as μg/m²(leaf)/h. And papers that we converted the values already mentioned in detail in the footnote of the Table 1.
Also, the emission unit can be μg/m²(leaf)/h, as seen in Pihlatie et al., 2005, and Zhu et al., 2022.

*Pihlatie, M., Ambus, P., Rinne, J., Pilegaard, K., and Vesala, T. (2005). Plant-mediated nitrous oxide emissions from beech (Fagus sylvatica) leaves. New Phytol. 168, 93–98. doi: 10.1111/j.1469-8137.2005.01542.x*
*Zhu C.F., Luo HD , Luo LC , Wang KY, Liao Y , Zhang S, Huang SS, Guo XM, Zhang L. Nitrogen and Biochar Addition Affected Plant Traits and Nitrous Oxide Emission From Cinnamomum camphora. Frontiers in Plant Science,13, 2022. 10.3389/fpls.2022.905537*

*Table 1 also has negative emissions for some species/periods. What are these? My first guess would be deposition, but then why aren't the species deposited at other stages? In any case, nothing is explained.*
**Reply:** We appreciate your insight regarding the negative emissions. These reflect depositions that are also related to local conditions and BVOC atmospheric concentrations. We will provide a detailed explanation in Section 2 to clarify this point. Thank you for highlighting this area that could benefit from further clarification.

● *Tables 2, 4. Same points as with Table 1.*
**Reply:** We understand that the distinction between emissions and emission factors might have been unclear. Our intention was to present emissions in Tables 2 and 4, as we aimed to highlight the

significance of species-specific emission for arable crops, bare soil, and cover crops. We will ensure this distinction is made clearer in the revised manuscript. We also re-precise that the objective is to show orders of magnitude of different emissions related to different soils, crops, phenological stages, etc.

●     *Table 5. Are these emission factors for 30C, 1000 μmole/m2/h, or for MEGAN2 conditions, or something else?*

**Reply:** The details mentioned by Reviewer 1 have already been addressed in Lines 300-301: "EFs for crop BVOC are currently applying the same standard conditions as those used for tree species (Guenther et al., 2012)." In this review paper, EFs for wheat and rapeseed were derived from measurements in France using inverse MEGAN v2.1, along with field environmental data inputs. Since EFs' standard conditions follows the description in Guenther et al., 2006 [MEGAN v2.0] (Guenther et al., 2012) and "the factor γ is equal to unity under these standard conditions (Guenther et al., 2006) [e.g., as described for isoprene below]," we did not modify any of the default standard conditions in the model.

*'The standard conditions for the MEGAN canopy-scale emission factors include a leaf area index, LAI, of 5 and a canopy with 80% mature, 10% growing and 10% old foliage; current environmental conditions including a solar angle (degrees from horizon to sun) of 60 degrees, a photosynthetic photon flux density (PPFD) transmission (ratio of PPFD at the top of the canopy to PPFD at the top of the atmosphere) of 0.6, air temperature=303 K, humidity=14 g kg−1 , wind speed=3 m s−1 and soil moisture=0.3 m3 m−3 ; average canopy environmental conditions of the past 24 to 240 h include leaf temperature=297 K and PPFD=200μmol m−2 s −1 for sun leaves and 50μmol m−2 s −1 for shade leaves.'*

●     *In Sect. 4.2.1, L296, it is stated that the standard conditions for EFs are 30C and 1000 μmole/m2/s, but MEGAN2 uses a much more complicated definition. I assume that Table 5 is for 30C, but as with other Tables, this is not explicit.*
**Reply:** 1. We understand that this is a bit ambiguous in the current writing of the paper, we will clarify and state clearly the conditions. For the standard conditions, we mentioned more details in the last response to Reviewer 1's comment above.
2. Yes, this is correct. All tables, except for Table 5, present emissions rather than emission factors. These emissions are not intended to be directly inputted into MEGAN or other models.

●     *Sect 3, L131 "Studies show...."? This important section makes statements about BVOC emissions, but no citations are given. Which studies? Table 3 is referred to, but no citations appear there. (In the footnote to this table there is further forwarding to different sections later in the text, which makes the table awkward to read. It would be better with a Table row giving such information.)*
**Reply**: Thanks for this comment. We will add an appendix table to show the references used in this table.

●     *The paper makes very little mention of the differences or issues surrounding leaf-scale versus canopy scale versus ecosystem scale emissions. Thus the sentence starting on L225 suddenly mentions that emissions may be reduced on an ecosystem scale, but no real explanation is given.*
**Reply:** We will revise the description to make the information from different scales clearer. Thank you for pointing out Line 225, we will add an explanation for the cited statement.

●     *Section 4.1 "Numerical modeling of BVOC" is also confusing. On L266 they define EFs as the "abundance" of a type of of gas/pollutant, but one would normally define EFs in mass released per unit leaf-area or leaf-mass per unit of time. The cited "Cheremisinoff 2011" paper isn't in the reference list, and I would anyway have expected a Guenther-type reference here. On L271 the paper states that "a uniform plant type is applied", but where, by whom? On L272-274 I am not sure what the link is between the Pierce statement and the Guenther 2013 reference.*
**Reply:** Thanks. We understand the confusing and hard to follow of this section due to the multiple model approaches. In the revised version, we will (1) Redefine EFs: describe in detail the parameters we imported for get EFs from MEGAN v2.1. Such as what we answered to Reviewer 1 about the EF

standard conditions. And for the EFs acquired from others' studies that already calculated by them, we will mention their based model and hypothesis from their paper, such as in Havermann et al. (2022), where SEF was derived from the Guenther model but applied the 'electron transport dependencies in the JJv' to the 'modifiers for light and temperature. (2) Also, we will rewrite the paragraph by separating each model and stating how agricultural land uses are treated (or not) in each.

● *The text is very qualitative, e.g. on L71 we read "emitted at relatively low rates", on L89 we have a "considerably higher emission rate". On L146 we read that "toluene is abundant in soil", but are there substantial emissions, also in comparison to e.g. road traffic emissions? Very much of the important section 3 is qualitative, making it difficult to know if emissions are really potentially important, or simply something somebody measured, somewhere. Similarly, on L237 we read that "a large amount of acetone..." is possible, but large compared to what?*
**Reply:** We will rewrite this paragraph by giving the numbers for each cited reference to make this section become more comparable.

● *L279-281 states that MEGAN2.1 has 19 VOC compounds for 15 plant categories; are these 285 EFs supported by measurements? How many are?! I would have hoped that a "review" of such BVOC emissions, and with Alex Guenther as coauthor, would have provided more background to such issues.*
**Reply:** 'MEGAN2.1 has 19 VOC from 15 plant categories' mentioned in Guenther et al 2012. The 19*15 = 285 EFs and can be found in the mentioned paper. We agree that it is good to have a critical view on the different models cited. We will try to dig in these EFs in a revised version of the manuscript. In any case, this review paper focuses solely on agricultural land; we do not provide information for other landscapes.

*In Section 4.2.2 I missed a discussion of the very real uncertainties associated with the specification of agricultural events: dates of sowing, emergence, growth, and fertilizer application. I know this is mentioned in the last paragraph, but the wording is rather vague. Is there any realistic hope of using satellite data to specify phenology and agricultural practices for European and/or global scale modeling? What would be needed to make progress in this field?*
**Reply:** In Section 4.2.2, we discussed the limitations of agricultural management-related data in lines 337 to 344. We did not suggest that 'there is no realistic hope of using satellite or statistical data.' On the contrary, we strongly encourage Reviewer 1 and other readers to consider using these data for larger-scale modeling. Here I attached some references about satellite images can (and are actually used) to estimate some agricultural practices. For example:

Dodin, M.; Smith, H.D.; Levavasseur, F.; Hadjar, D.; Houot, S.; Vaudour, E. Potential of Sentinel-2 Satellite Images for Monitoring Green Waste Compost and Manure Amendments in Temperate Cropland. *Remote Sens.* 2021, *13*, 1616. https://doi.org/10.3390/rs13091616

Veloso, Amanda, Stéphane Mermoz, Alexandre Bouvet, Thuy Le Toan, Milena Planells, Jean-François Dejoux, et Eric Ceschia. 2017. « Understanding the temporal behavior of crops using Sentinel-1 and Sentinel-2-like data for agricultural applications ». Remote Sensing of Environment 199 (septembre):415-26. https://doi.org/10.1016/j.rse.2017.07.015.

Wijmer, T., Al Bitar, A., Arnaud, L., Fieuzal, R., and Ceschia, E.: AgriCarbon-EO: v1.0.1: Large Scale and High Resolution Simulation of Carbon Fluxes by Assimilation of Sentinel-2 and Landsat-8 Reflectances using a Bayesian approach, EGUsphere [preprint], https://doi.org/10.5194/egusphere-2023-48, 2023

And it is still ongoing research, and other methods are also possible like national databases such as :

Levavasseur, F., P. Martin, C. Bouty, A. Barbottin, V. Bretagnolle, O. Thérond, O. Scheurer, et N. Piskiewicz. 2016. « RPG Explorer: A new tool to ease the analysis of agricultural landscape dynamics with the Land Parcel Identification System ». Computers and Electronics in Agriculture 127 (septembre):541-52. https://doi.org/10.1016/j.compag.2016.07.015.

However, as a review paper, we also have the responsibility to highlight potential challenges for future use, so that scientists in related fields can address and improve upon them. Considering that other audiences may share the same concerns as Reviewer 1, we can consult with agronomists and incorporate their current solutions into the corresponding section for a more detailed discussion.

● *Section 5.1 (L357) starts "MEGAN was applied as a base model (Eq. 5.1) to estimate BVOC emissions from agricultural herbaceous crops...", but MEGAN isn't applied here. Further, if I understood right Table 5 gives emission factors using the older 30 deg C definition of emission factors, whereas MEGAN requires much more complex conditions.*

**Reply:** 1. MEGAN v2.1 (Guenther et al., 2012) includes a PFT15 function specifically for crops.

2. The emission factor (EF) and canopy model for light and radiation distribution will be modified. In lines 358-359, a portion of the sentence was omitted, specifically the phrase 'the canopy model for LAI depth distribution.' As a result, only two citations related to crop-specific growth formats were left in that section.

3. We appreciate your perspective regarding the '30°C' statement. In line 301, we aimed to clarify that our approach involves inverting MEGAN, considering all relevant meteorological and environmental inputs. We will revise this section to ensure that the process and reasoning are fully transparent and accurately presented.

● *Section 5.1 continues (L358) to say "we modified EFs to crop/grass species-dependent values", but no details are given of the resulting EFs. This is all very confusing!*

**Reply:** We propose to use MEGAN as a base model. EFs will acquire by applying the environmental measurements, include air temperature (for leaf temperature), relative humidity, air pressure, PPFD, soil moisture, wind speed, and LAI, then inverse model version 2.1 to get the best EF to match the net fluxes from measurement.

● *Section 5.3 was also wordy but vague; which information here can be used, and/or what is needed before we can use such information.*

**Reply:** Thank you. We will separate this paragraph into two sections: 1. recommended practices that should be accounted in future models and for which there are some existing measurements in the literature (e.g. different type of organic fertilization applied (Abis et al., 2020, 2021; Haider et al., 2022)).

2. Few hypotheses concerning agricultural practices that could have an impact on BVOC emissions and should be further investigated, e.g. tillage, irrigation.

● *Section 6, Conclusions states the paper presents "a table of emissions during different phenological stages", and that they "provide a list of crop-specific emission factors for dominant BVOCs", but as noted above the numbers provided are confusing and probably not useful.*

**Reply:** Thank you. This will be clarified as stated above in response to your specific comment. We will also mention this statement as potential research points for further agricultural VOC measurement approaches.

● *Data availability: these days data should be provided in SI, or via zenodo I think. Available from the authors on request is always dependent on the availability of the authors.*

**Reply:** Thank you for this comment. We did not expect the reviewer to interpret this statement in this way. It is easy to arrange data from a research paper because all data are measured by the authors or their team. However, as a review paper, we have data from other groups, which means we have the right to use it but must respect their work. Therefore, we state here that the data is available from the authors upon request.

*Smaller points:*

*The English needs a thorough revision. There are many cases where cases don't match (leaf versus leaves for example), and some cases where the sentences don't make sense (e.g. L82-84). Other examples: L86 - I guess you mean emergence and not emergency? ; L109: what does "besides the fate of VOCs" mean here? ; L140: "and promote new compounds show" isn't English; L150: Does "a positive response" mean increased emission rate? ; L166 delete "to" from "to organic waste"*

*Table 1: it would be useful for non-agriculturalists to give the approximate time-periods of the different stages.*

**Reply:** We appreciate your observation and agree that this is an important point. We will make the necessary revisions to improve clarity. We will seek help from a native English speaker to review the language.

*L96. The paper states that "The soil continues emitting BVOCs during plant growth and ripening, but emission rates from this period have not been reported so far to our knowledge." So, how do you know that BVOCs are still emitted?*

**Reply**: We mentioned that soil continues to emit VOCs even when plants are present in the field because: 1. Microbial activity, soil moisture, and particle absorption are primary contributors to soil VOC emissions (e.g., Tang et al., 2019), which depend on soil properties and the amount of soil organic matter. Plants, particularly their residues and root systems, can increase soil organic matter content to some extent.

2. Agricultural soil measurements provide insights under specific conditions, such as during a heatwave with wheat in the field, as observed by Schade et al., 2004 (https://doi.org/10.1016/j.atmosenv.2004.08.017). We can add these references in the corresponding section and clearly state the specific conditions and influencing factors.

*L141. The last sentence is so vague ("Variations of BVOC..") as to be meaningless.*
**Reply:** Thanks for the comment. We will revise accordingly.

*L217-220. This is a bit vague and unclear. If drought reduces BVOC emissions, wouldn't one get less secondary organic aerosol, not more?*
**Reply:** In line 213 - 216, we already started to talk about increase effects rather than reduce effects. The chemistry behind SOA formation is complex and less emissions does not necessarily imply less SOA formation (Bonn et al., 2019; Wang et al., 2023).
*Wang, Ruipeng, Wenjiao Duan, Shuiyuan Cheng, et Xiaoqi Wang. 2023. Nonlinear and lagged effects of VOCs on SOA and O3 and multi-model validated control strategy for VOC sources ». Science of The Total Environment 887 (08):164113. https://doi.org/10.1016/j.scitotenv.2023.164113.*

*L223. The text here and around relies a lot on Bonn et al., 2019, but that paper only dealt with trees. Also, many monoterpene emissions are not under stomatal control, being rather stored in pools within the leaf (e.g. Niinemets et al., 2004, Guenther et al - many papers).*
**Reply:** Bonn et al., 2019 was cited in the paragraph discussing how stressed VOCs contribute to particle formation L217-220. We did not cite this paper in the section that discusses the types of VOCs emitted by crops during drought. However, we will clarify this distinction for the readers and add references specific to that.

*L247. The statement "Cover crops are planted a few months between two main crops" is likely true in France, but do all countries have two main crops, with a few months between them? In general, this paper has little consideration of climatological differences between even parts of Europe, let alone the globe.*
**Reply:** We will narrow and state the regional information in our title and material section.
About cover crop, *'is likely true in France, but do all countries have two main crops'*, see reference: e.g. Fendrich et al., 2023.
*Arthur Nicolaus Fendrich, Francis Matthews, Elise Van Eynde, Marco Carozzi, Zheyuan Li, Raphael d'Andrimont, Emanuele Lugato, Philippe Martin, Philippe Ciais, Panos Panagos. From regional to parcel scale: A high-resolution map of cover crops across Europe combining satellite data with statistical surveys, Science of The Total Environment, 873, 2023, 162300, https://doi.org/10.1016/j.scitotenv.2023.162300.*

Although cover crop is applied in short period, and normally during winter time, but as an approach of management practice with less focusing, we believe it worthy and should be mentioned in our review paper. In the revised version, we will only focuses (as stated in the reply before as well) on Europe and North American conditions.

*L262: Usually the "and" can be dropped between chemistry and transport models.*

*L263. Give MEGAN a proper reference.*

*L288 states "To our knowledge" about MEGAN and LPJ-GUESS, but you have the lead author of MEGAN on the author list. I am sure Alex Guenther knows. And it would not take much effort to ask the LPJ developers about the EFs being discussed.*

*L323: What is "airflow" emitted?*

**Reply:** Thanks for the comment. We will revise accordingly.

*L371: why are natural environments relevant here. Agricultural land us far from natural.*

**Reply:** The wording here is confusing this will be clarified.

*L365 on, Section 5.2. Is WFPS a good indicator of soil moisture? Soils with the same WFPS can have very different soil water pressure values.*

**Reply:** WFPS is a mixed variable informing about soil water content but also $O_2$ availability in the soil. We believe it is a good driver for microbial activity.

*L395-397 - this text is unclear. Re-word.*

*General: why are italics used for words such as "dry weight" in Table 6, and in units throughout the text, e.g., for g, kg, m or h? Also, g and kg and VOC should not be italic.*

**Reply:** Thanks for the comment. We will revise accordingly.